# Drop or Merge? Hybrid MoE LLMs Compressors via Metric-Driven Adaptive Allocation

## Abstract

Mixture-of-Experts (MoE) models enhance the scalability of large language models but encounter deployment challenges due to their vast parameter counts. Existing compression methods either drop experts entirely (discarding valuable knowledge) or merge experts (suffering from parameter conflicts), typically employing uniform strategies that ignore the heterogeneous specialization patterns across layers. In this paper, we propose DM-MoE, an adaptive Drop-then-Merge MoE compression framework to address these limitations. Our approach is motivated by two key observations: first, that eliminating a small number of truly redundant experts facilitates more effective subsequent merging, and second, that expert functional redundancy and behavioral similarity serve as reliable indicators for adaptive compression throughout MoE architectures. Building on these insights, we develop a two-stage compression: (1) In the dropping phase, we quantify layer redundancy via mutual information between expert outputs and formulate a constrained optimization problem to derive layer-wise dropping budgets, then select experts based on output impact assessment to retain those with high functional significance. (2) In the merging phase, we adaptively determine the number of expert groups per layer using behavioral diversity metrics, partition experts into functionally similar clusters via graph-based optimization, and merge them using importance-weighted averaging based on activation frequency and output deviation. Comprehensive evaluations on Mixtral, Qwen, DeepSeek and GPT-OSS MoE demonstrate that our DM-MoE surpasses state-of-the-art methods across models and compression ratios. For Mixtral-8×7B, we retain 96.5%/89.1% of original performance at 25%/50% expert reduction. Code is available in the Appendix.

## 1 Introduction

Large Language Models (LLMs) have revolutionized natural language processing (OpenAI et al., 2024; Team et al., 2024), with Mixture-of-Expert (MoE) architectures emerging as a particularly promising approach for achieving state-of-the-art performance while improving computational efficiency (Jiang et al., 2024; Team, 2024). By conditionally activating only a subset of model parameters for each input, MoE architectures can achieve superior performance compared to dense models of equivalent computational cost (Dai et al., 2024). Despite these efficiency advantages, MoE LLMs still present substantial deployment challenges, particularly due to their enormous parameter counts. This parameter explosion leads to prohibitive storage requirements, increased memory bandwidth demands, and higher serving costs in production environments (Imani et al., 2024).

To compress the parameter size of MoE LLMs, expert dropping methods (Lu et al., 2024; Muzio et al., 2024; Yang et al., 2024b) identify and remove less important or redundant experts based on various criteria such as activation frequency, importance scores, or contribution to output. These approaches include regularization-based techniques (Chen et al., 2022; Muzio et al., 2024) that penalize certain experts during fine-tuning, search-based methods (Lu et al., 2024; Yang et al., 2024b) that evaluate different expert subsets, and heuristic approaches based on pre-defined metrics (He et al., 2024). Recent expert merging strategies consolidate multiple experts into fewer, merged representations through techniques like weighted averaging. Among these, MC-SMoE (Li et al., 2023a), HC-SMoE (Chen et al., 2024), and EEP (Liu et al., 2024) employ distinct fusion criteria: frequency-based selection, hierarchical clustering, and search-based optimization, respectively.

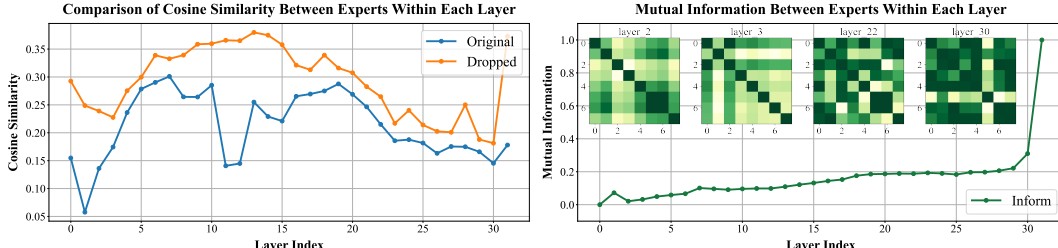

Figure 1: **Left:** Expert parameter alignment degree (measured by cosine similarity) between merged experts and their source experts, comparing merging from the original full expert set versus merging from the reduced expert set (after 25% expert reduction) using average merging on the Mixtral-8×7B. **Right:** intra-layer expert mutual information matrices of layer [2, 3, 22, 30] on Mixtral-8×7B and information values for different layers. More analyses are available in Appendix B.1.

**Problem Statement:** However, these expert dropping/merging methods suffer from several critical limitations: **(1) Performance collapse from complete dropping:** Expert dropping methods fundamentally discard portions of the model's learned knowledge. At higher compression ratios, this knowledge loss becomes particularly problematic, leading to significant performance degradation that often requires expensive fine-tuning to recover. The complete removal of experts creates representation gaps that the remaining network struggles to compensate for, especially in specialized tasks where the dropped experts may have encoded critical domain knowledge. **(2) Parameter conflicts in direct merging:** In well-trained MoE models, experts naturally specialize into distinct functional roles with potentially orthogonal parameter distributions. When experts with diverse specializations are directly merged, the resulting consolidated experts often suffer from destructive interference between conflicting parameters. This parameter averaging dilutes the specialized capabilities of the constituent experts, creating compromised representations that inadequately capture the functional diversity of the original expert set. **(3) Uniform compression ignoring layer sensitivity:** most drop/merge approaches apply uniform compression across all layers, overlooking varying sensitivity patterns. Recent studies (Li et al., 2024) reveal significant variation in expert redundancy across layers, with early layers requiring preferential treatment. Some expert dropping methods explore adaptive ways but require time-consuming evolutionary search processes (Liu et al., 2024).

**Our New Observations and Framework:** In this paper, we introduce DM-MoE, a novel compression framework that addresses these limitations through a sequential drop-then-merge paradigm. Our framework is motivated by two key observations from our analysis of expert behavior in MoE models: **(1) Strategic dropping facilitates effective merging:** Figure 1 (*left*) demonstrates that dropping 25% of unimportant experts first allows the remaining fewer experts to achieve higher parameter alignment with the final merged expert across all MoE layers. This strategic pre-dropping reduces parameter conflicts among the experts to be merged, resulting in merged experts that maintain better parameter consistency with their source experts compared to merging the original full expert set. **(2) Expert metrics reveal hierarchical specialization patterns:** As shown in Figure 1 (*right*), we observe that mutual information metrics precisely capture MoE layer-wise sensitivity: early layers maintain low mutual information, indicating high specialization requiring preservation; later layers show progressively increasing mutual information, exhibiting redundancy amenable to aggressive compression. Building on these insights, our DM-MoE represents a two-phase compression framework that sequentially drops and merges experts. **In the first phase,** we perform layer-wise adaptive expert dropping guided by information-theoretic metrics. We use Canonical Correlation Analysis (CCA) to measure mutual information between expert outputs, quantifying functional redundancy within each layer. This enables us to allocate layer-specific retention budgets through constrained optimization: layers with irrelevant experts retain more, while redundant layers undergo aggressive pruning. Within each layer, we select experts to keep based on their output impact, preserving those whose removal would most affect the layer's functionality. **In the second phase,** we employ a graph-based, layer-wise strategy to merge experts. We begin by modeling each layer as a similarity graph, where edges quantify the behavioral correlation between experts. Our process first involves an inter-layer allocation step to determine the optimal number of expert groups for each layer, assigning more groups to layers with greater diversity. We then partition the graph for each layer to form expert groups by maximizing intra-group similarity, which ensures coherent merging. Finally, we merge experts within each layer using a dual-weighted factor that combines activation

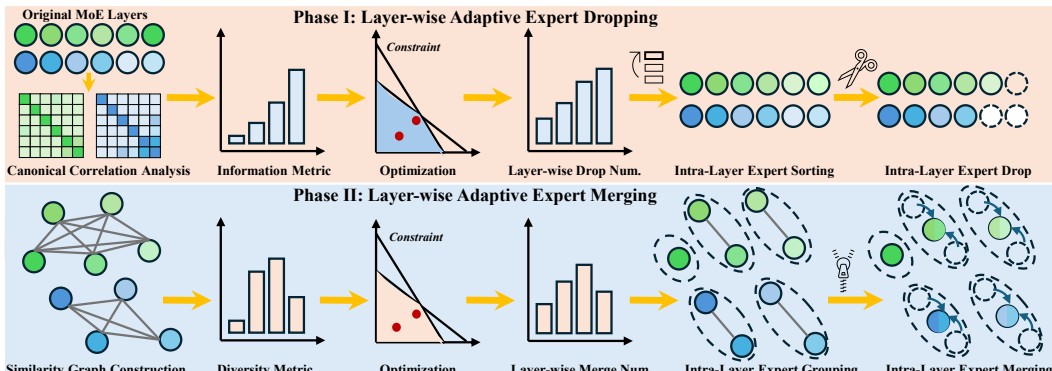

Figure 2: Overview of our DM-MoE framework, comprising two core phases: **(1) Expert Dropping:** we use information metrics from canonical correlation analysis to optimize layer-wise drop counts, then perform intra-layer expert sorting and selective dropping; and **(2) Expert Merging:** we allocate layer-specific merging groups via constrained optimization using diversity metrics derived from graph construction, then apply intra-layer expert grouping via graph partitioning and merging.

frequency and output deviation scores. This dual-factor merging preserves critical functionalities, effectively mitigating parameter conflicts.

**Evaluation Results:** Our comprehensive evaluation across five MoE models (Mixtral-8×7B, Qwen1.5-MoE-A2.7B, Qwen3-30B-A3B, DeepSeek-V2-Lite, and GPT-OSS-20B) demonstrates the superior performance of DM-MoE across diverse compression scenarios. Our DM-MoE retains 96.5% of Mixtral's original accuracy with 25% fewer experts, surpassing HC-SMoE (Chen et al., 2024) by +4.3% and Frequency-drop by +6.3%. Notably, even under aggressive 50% compression, it achieves 89.1% retention for Mixtral-8×7B, 85.9% for Qwen1.5-MoE, significantly outperforming prior methods. For 50% compression on recent models like Qwen3-30B-A3B, DeepSeek-V2-Lite, and GPT-OSS-20B, DM-MoE maintains 81.2%, 75.5%, and 83.2% of original performance respectively, demonstrating consistent advantages of 7-18% over the strongest baseline.

**In summary, our contributions are threefold:**

**(1)** Based on the observation that dropping unimportant experts mitigates parameter conflicts, we propose DM-MoE, an adaptive Drop-then-Merge paradigm that **aims at memory footprint and parameter size reduction** while preserving MoE performance.

**(2)** We introduce a new adaptive allocation scheme driven by dual metrics of information and similarity. By formulating the allocation as a linear optimization problem, DM-MoE flexibly captures the hierarchical characteristics of MoE architectures.

**(3)** We conduct extensive experiments across diverse MoE architectures, including Mixtral, Qwen, DeepSeek, and GPT-OSS. The results demonstrate that DM-MoE consistently outperforms state-of-the-art expert reduction and merging methods.

## 2 DM-MoE: ADAPTIVE DROP-THEN-MERGE MoE COMPRESSION

Our overall process is illustrated in Figure 2. Given an MoE model with $L$ layers and $N$ experts per layer, our goal is to compress it while minimizing performance degradation. Our approach first identifies and removes less important experts, reducing the expert count per layer from $N$ to $K_l$ based on layer-specific importance metrics; subsequently, an expert merging phase further consolidates the remaining experts into $G_l$ merged groups (where $G_l \leq K_l$) through similarity-based clustering, effectively addressing the parameter conflict issues that plague direct merging approaches.

### 2.1 PHASE I: LAYER-WISE ADAPTIVE EXPERT DROPPING

Our expert dropping phase involves creating information metrics, allocating drop counts per layer, and ranking experts to remove the unimportant ones.

**Information-Aware Metric Construction.** To accurately capture the hierarchical redundancy characteristics within MoE layers, we employ Canonical Correlation Analysis (CCA) to estimate pairwise mutual information between expert outputs (Kornblith et al., 2019). Unlike simple correlation measures, CCA reveals the full spectrum of linear dependencies between high-dimensional expert representations, making it particularly suitable for identifying redundant layers (Li et al., 2023b). For $N$ experts in layer $l$, the MoE computation processes an input token $\mathbf{x} \in \mathbb{R}^d$ through expert modules $\{E_1, E_2, \ldots, E_N\}$ and router $R$ to produce output $\mathbf{y} = \sum_{i=1}^{N} p_i(\mathbf{x}) \cdot E_i(\mathbf{x})$, where $p_i(\mathbf{x})$ denotes routing probability and $E_i(\mathbf{x})$ represents expert output. The router employs top-$k$ gating with softmax normalization, activating only the $k$ most relevant experts per token to maintain computational efficiency. For each expert pair $(E_i, E_j)$, we collect their outputs $\mathbf{Y}_i \in \mathbb{R}^{m \times d}$ and $\mathbf{Y}_j \in \mathbb{R}^{m \times d}$ over $m$ calibration samples. CCA identifies linear projections that maximize correlation between these representations by computing canonical correlations $\{\rho_k\}_{k=1}^{d}$ as singular values of:

$$\mathbf{T} = \mathbf{C}_{ii}^{-1/2} \mathbf{C}_{ij} \mathbf{C}_{jj}^{-1/2}, \tag{1}$$

where $\mathbf{C}_{ij}$ represents the cross-covariance matrix between expert outputs. The mutual information between experts is estimated as:

$$I_{ij} = -\frac{1}{2} \sum_{k=1}^{d} \log(1 - \rho_k^2). \tag{2}$$

The layer-wise total mutual information aggregates all pairwise values: $I_{\text{layer}} = \sum_{i=1}^{N-1} \sum_{j=i+1}^{N} I_{ij}$. We apply sigmoid normalization to obtain the information score:

$$D_{\text{info}} = 1 - \frac{1}{1 + e^{-I_{\text{layer}}}}, \tag{3}$$

Based on the principle that higher mutual information indicates greater functional redundancy among experts, the final expert score $D_{\text{info}}$ inversely correlates with mutual information: layers with lower mutual information receive higher scores, indicating their experts have more distinctive functional roles that should be preserved during compression.

**Inter-Layer Expert-Drop Allocation.** We formulate layer-wise expert retention as a constrained optimization problem that maximizes preserved diversity:

$$\max_{K_1, \ldots, K_L} \quad \sum_{l=1}^{L} D_{\text{info}}^l \cdot \phi(K_l)$$

$$\text{subject to:} \quad \sum_{l=1}^{L} K_l = K_{\text{total}} \tag{4}$$

$$K_{\min} \leq K_l \leq N \quad \forall l \in \{1, \ldots, L\}$$

$$|K_l - K_{l+1}| \leq \Delta_{\max} \quad \forall l \in \{1, \ldots, L-1\},$$

where $K_l$ denotes experts retained in layer $l$, $K_{\text{total}}$ represents the global retention budget, and transformation function $\phi(K_l)$ captures diminishing returns of additional experts. The smoothness constraint $\Delta_{\max}$ limits expert count differences between adjacent layers, preventing abrupt capacity changes that could disrupt information flow through the LLMs. This constrained optimization can be efficiently handled by the SLSQP solver in scipy.optimize Gommers et al. (2024), usually converging in under 0.5 seconds for MoE LLMs with 30–80 layers (see Appendix Table 14).

**Intra-Layer Expert Dropping.** Once the retention budget $K_l$ for each layer is determined, we select which specific experts to keep using an output impact assessment approach that identifies experts whose removal would minimally affect the layer's functionality. For each expert $E_i$ in layer $l$, we measure the output deviation when that expert is removed:

$$\delta_i = \frac{1}{|\mathcal{X}|} \sum_{\mathbf{x} \in \mathcal{X}} \|\mathbf{y}_l(\mathbf{x}) - \mathbf{y}_l^{-i}(\mathbf{x})\|_2, \tag{5}$$

where $\mathbf{y}_l(\mathbf{x})$ is the original layer output for input $\mathbf{x}$, and $\mathbf{y}_l^{-i}(\mathbf{x})$ is the layer output with expert $E_i$ removed and routing weights redistributed among remaining experts. We then select the $K_l$ experts

with the largest output deviation scores, as these experts have the most significant impact on the layer's behavior. To address the computational complexity of output perturbation in this process, we employ greedy search within a smaller candidate pool based on statistical information metrics averaged across experts. This approach effectively preserves the most functionally significant experts while discarding those whose contribution can be compensated by other experts in the same layer.

## 2.2 PHASE II: LAYER-WISE ADAPTIVE EXPERT MERGING

Following the expert dropping phase, we introduce a graph-based layer-wise merging strategy that fundamentally reimagines how we understand and exploit expert relationships within MoE layers. We observe that experts in MoE models naturally form complex relational structures that traditional merging methods fail to capture adequately. We conceptualize each MoE layer as a fully connected graph $\mathcal{G}^l = (\mathcal{V}^l, \mathcal{E}^l, \mathbf{W}^l)$, where vertices $\mathcal{V}^l$ represent the $K_l$ remaining experts after dropping, edges $\mathcal{E}^l$ encode pairwise similarities, and weights $\mathbf{W}^l$ quantify that expert similarities.

**Similarity Graph Construction.** For each layer $l$, we construct the expert similarity graph by collecting output representations from all remaining experts. We compute the similarity weight between experts $E_i$ and $E_j$ as:

$$w_{ij}^l = \frac{\langle \mathbf{y}_i, \mathbf{y}_j \rangle}{\|\mathbf{y}_i\| \|\mathbf{y}_j\|}, \tag{6}$$

where $\mathbf{y}_i$ and $\mathbf{y}_j$ represent the output activations of experts $E_i$ and $E_j$ respectively, and $\langle \cdot, \cdot \rangle$ denotes the inner product. We quantify the behavioral diversity of layer $l$ through:

$$D_{\text{div}}^l = -\frac{2}{K_l(K_l - 1)} \sum_{i=1}^{K_l-1} \sum_{j=i+1}^{K_l} w_{ij}^l, \tag{7}$$

where larger values of $D_{\text{div}}^l$ indicate greater behavioral diversity among experts.

**Inter-Layer Expert-Merge Allocation.** We formulate the allocation of expert groups across layers as a linear program that optimizes the distribution based on diversity metrics. We solve for all layers simultaneously:

$$\max_{G_1,\ldots,G_L} \quad \sum_{l=1}^{L} D_{\text{div}}^l \cdot \phi(G_l)$$

$$\text{subject to:} \quad \sum_{l=1}^{L} G_l = G_{\text{total}} \tag{8}$$

$$1 \leq G_l \leq K_l \quad \forall l \in \{1, \ldots, L\}$$

$$|G_l - G_{l+1}| \leq \Delta_{\max} \quad \forall l \in \{1, \ldots, L-1\},$$

where $G_{\text{total}}$ denotes the total number of expert groups after merging, and $\Delta_{\max}$ controls the smoothness of allocation across adjacent layers. This formulation ensures that layers with higher diversity retain more expert groups, preserving their functional richness.

**Graph Partitioning for Expert Grouping.** Having determined the optimal number of groups $G_l$ for each layer, we partition the similarity graph $\mathcal{G}^l$ to assign experts to groups. Unlike hierarchical clustering's irrevocable local decisions or K-means' spherical cluster assumptions, we formulate a global optimization problem (Çatalyürek et al., 2023) that partitions $K_l$ experts into $G_l$ disjoint groups:

$$\max_{\mathcal{P}^l} \quad \sum_{k=1}^{G_l} \sum_{i,j \in V_k, i<j} w_{ij}^l$$

$$\text{subject to:} \quad \bigcup_{k=1}^{G_l} V_k = \mathcal{V}^l, \quad V_i \cap V_j = \emptyset \ \forall i \neq j, \tag{9}$$

where $\mathcal{P}^l = \{V_1, V_2, \ldots, V_{G_l}\}$ represents the partition. This formulation maximizes intra-group similarity by considering all expert relationships simultaneously, avoiding the local decision pitfalls of hierarchical methods. The resulting partitions create more coherent expert groups that minimize information loss during merging and better preserve the model's original capabilities.

Table 1: Results of our DM-MoE and HC-SMoE in three recent MoE LLMs. We report accuracy (higher is better↑) on eight diverse reasoning and understanding tasks.

| Expert | Method | ARC-c | ARC-e | BoolQ | HellaS. | MMLU | OBQA | RTE | WinoG. | Average↑ |
|---|---|---|---|---|---|---|---|---|---|---|
| | | | | | **Qwen3-30B-A3B** | | | | | |
| Num=128 | Original | 0.534 | 0.797 | 0.888 | 0.596 | 0.778 | 0.352 | 0.827 | 0.710 | 0.685 |
| Num=96 | HC-SMoE | 0.349 | 0.637 | 0.822 | 0.401 | 0.549 | 0.220 | 0.733 | 0.613 | 0.540 |
| | **DM-MoE (Ours)** | **0.481** | **0.765** | **0.869** | **0.543** | **0.666** | **0.292** | **0.841** | **0.696** | **0.644** |
| Num=64 | HC-SMoE | 0.229 | 0.438 | 0.634 | 0.292 | 0.298 | 0.132 | 0.500 | 0.498 | 0.378 |
| | **DM-MoE (Ours)** | **0.398** | **0.675** | **0.817** | **0.446** | **0.5035** | **0.276** | **0.711** | **0.620** | **0.556** |
| | | | | | **DeepSeek-V2-Lite** | | | | | |
| Num=64 | Original | 0.455 | 0.769 | 0.727 | 0.550 | 0.497 | 0.320 | 0.617 | 0.673 | 0.576 |
| Num=48 | HC-SMoE | 0.370 | 0.705 | 0.677 | 0.460 | 0.292 | 0.288 | 0.567 | 0.665 | 0.503 |
| | **DM-MoE (Ours)** | **0.378** | **0.709** | **0.685** | **0.499** | **0.389** | **0.292** | **0.599** | **0.686** | **0.530** |
| Num=32 | HC-SMoE | 0.281 | 0.576 | 0.587 | 0.362 | **0.240** | 0.190 | 0.505 | 0.587 | 0.416 |
| | **DM-MoE (Ours)** | **0.301** | **0.604** | **0.617** | **0.369** | 0.231 | **0.202** | **0.560** | **0.597** | **0.435** |
| | | | | | **GPT-OSS-20B** | | | | | |
| Num=32 | Original | 0.453 | 0.774 | 0.757 | 0.415 | 0.566 | 0.270 | 0.679 | 0.658 | 0.571 |
| Num=24 | HC-SMoE | 0.294 | 0.574 | 0.619 | 0.340 | 0.417 | 0.200 | 0.639 | 0.624 | 0.463 |
| | **DM-MoE (Ours)** | **0.383** | **0.712** | **0.740** | **0.389** | **0.511** | **0.234** | **0.668** | **0.629** | **0.533** |
| Num=16 | HC-SMoE | 0.222 | 0.468 | 0.608 | 0.322 | 0.352 | 0.172 | 0.560 | 0.567 | 0.409 |
| | **DM-MoE (Ours)** | **0.301** | **0.634** | **0.685** | **0.344** | **0.367** | **0.208** | **0.682** | **0.577** | **0.475** |

**Intra-layer Expert Merging.** After obtaining the expert partitions, we merge experts within each partition by considering both their activation frequency and output deviation scores to model the importance of each expert. This dual-metric approach captures both the usage patterns (how often an expert is selected) and functional significance (how much the expert contributes to the layer's output), providing a more comprehensive assessment of expert importance than either metric alone.

For each expert $E_i$, we compute its importance weight as:

$$\alpha_i = \bar{f}_i + \bar{\delta}_i, \tag{10}$$

where $\bar{f}_i$ is the normalized activation frequency of expert $i$ and $\bar{\delta}_i$ is the normalized output deviation score computed earlier. This combination ensures that both frequently activated experts and those with high functional impact contribute more significantly to the merged representation.

For experts within the same partition $V_k$, we create a merged expert by computing the weighted average of their parameters:

$$\mathbf{W}^k_{\text{merged}} = \frac{\sum_{i \in V_k} \alpha_i \cdot \mathbf{W}_i}{\sum_{i \in V_k} \alpha_i}. \tag{11}$$

where $\mathbf{W}_i$ represents the parameters of expert $i$. This importance-weighted merging tends to preserve the most critical functionalities within each group while approximating the essential behavioral patterns of the original experts, creating merged experts that inherit collective capabilities proportional to individual importance.

Through these two phases, we strike a flexible balance between removing redundant experts and merging important ones. Our metric-driven optimization enables efficient adaptive allocation, completing core processing steps in about 10 minutes (see Appendix C.3) while avoiding expensive search.

## 3 EXPERIMENTS

### 3.1 EXPERIMENTAL SETUPS

We conduct experiments on cutting-edge MoE models: Qwen3-30B-A3B (Yang et al., 2024a), DeepSeek-V2-Lite (Dai et al., 2024), GPT-OSS-20B (Agarwal et al., 2025) Mixtral 8x7B (Jiang et al., 2024) and Qwen1.5-MoE-A2.7B (Team, 2024). We evaluate our method on eight diverse reasoning and understanding tasks (Gao et al., 2023) (*e.g.*, ARC (Clark et al., 2018), MMLU (Hendrycks et al., 2021)). We construct a calibration dataset of 16 sequences (2,048 tokens each) sampled from C4 for both the expert dropping and merging phases. Our compression budget is allocated equally

Table 2: Comparisons of MoE compression methods across different models and compression ratios. Frequency/output-drop baseline sorts and drops unimportant experts based on each expert's frequency/output within each MoE layer. We report accuracy (higher is better↑) on eight diverse reasoning and understanding tasks.

| Expert | Method | ARC-c | ARC-e | BoolQ | HellaS. | MMLU | OBQA | RTE | WinoG. | Average↑ |
|--------|--------|-------|-------|-------|---------|------|------|-----|--------|----------|
| | | | | | Mixtral-8×7B | | | | | |
| Num=8 | Original | 0.565 | 0.842 | 0.851 | 0.649 | 0.671 | 0.350 | 0.711 | 0.759 | 0.675 |
| Num=6 | Frequency-drop | 0.478 | 0.781 | 0.781 | 0.568 | 0.469 | 0.322 | 0.552 | 0.754 | 0.588 |
| | Output-drop | 0.468 | 0.772 | 0.750 | 0.576 | 0.464 | 0.298 | 0.599 | 0.751 | 0.585 |
| | MC-SMoE | 0.286 | 0.595 | 0.591 | 0.431 | 0.253 | 0.200 | 0.527 | 0.600 | 0.435 |
| | HC-SMoE | 0.450 | 0.730 | 0.830 | 0.570 | 0.560 | 0.290 | 0.690 | 0.745 | 0.608 |
| | **DM-MoE (Ours)** | **0.522** | **0.819** | **0.843** | **0.615** | **0.631** | **0.324** | **0.700** | **0.756** | **0.651** |
| Num=4 | Frequency-drop | 0.215 | 0.386 | 0.598 | 0.364 | 0.238 | 0.142 | 0.531 | 0.533 | 0.376 |
| | Output-drop | 0.214 | 0.392 | 0.628 | 0.384 | 0.237 | 0.164 | 0.538 | 0.556 | 0.389 |
| | MC-SMoE | 0.207 | 0.278 | 0.524 | 0.279 | 0.255 | 0.108 | 0.498 | 0.516 | 0.333 |
| | HC-SMoE | 0.322 | 0.613 | 0.754 | 0.493 | 0.392 | 0.256 | 0.614 | 0.671 | 0.514 |
| | **DM-MoE (Ours)** | **0.443** | **0.744** | **0.839** | **0.556** | **0.539** | **0.288** | **0.686** | **0.714** | **0.601** |
| | | | | | Qwen1.5-MoE-A2.7B-Chat | | | | | |
| Num=60 | Original | 0.396 | 0.705 | 0.812 | 0.593 | 0.598 | 0.312 | 0.737 | 0.658 | 0.601 |
| Num=45 | Frequency-drop | 0.327 | 0.568 | 0.766 | 0.547 | 0.426 | 0.290 | 0.729 | 0.648 | 0.538 |
| | Output-drop | 0.336 | 0.593 | 0.706 | 0.518 | 0.480 | 0.270 | 0.661 | 0.594 | 0.520 |
| | MC-SMoE | 0.371 | 0.646 | 0.755 | 0.531 | 0.383 | 0.252 | 0.776 | 0.673 | 0.548 |
| | HC-SMoE | 0.344 | 0.663 | 0.753 | 0.527 | 0.499 | 0.282 | 0.704 | 0.610 | 0.548 |
| | **DM-MoE (Ours)** | **0.354** | **0.615** | **0.802** | **0.525** | **0.59** | **0.252** | **0.733** | **0.659** | **0.566** |
| Num=30 | Frequency-drop | 0.261 | 0.413 | 0.616 | 0.388 | 0.246 | 0.198 | 0.545 | 0.569 | 0.405 |
| | Output-drop | 0.270 | 0.511 | 0.645 | 0.402 | 0.326 | 0.194 | 0.549 | 0.538 | 0.429 |
| | MC-SMoE | 0.189 | 0.326 | 0.568 | 0.287 | 0.231 | 0.176 | 0.448 | 0.524 | 0.344 |
| | HC-SMoE | 0.246 | 0.503 | 0.636 | 0.334 | 0.349 | 0.190 | 0.500 | 0.570 | 0.416 |
| | **DM-MoE (Ours)** | **0.315** | **0.563** | **0.739** | **0.434** | **0.515** | **0.242** | **0.718** | **0.603** | **0.516** |

Table 3: Results of drop and merge settings via uniform/adaptive allocation for Mixtral 8×7B→4×7B.

| Method | ARC-c | ARC-e | BoolQ | HellaS. | MMLU | OBQA | RTE | WinoG. | Average↑ |
|--------|-------|-------|-------|---------|------|------|-----|--------|----------|
| Drop Only (uniform) | 0.432 | 0.723 | 0.759 | 0.536 | 0.403 | 0.288 | 0.585 | 0.717 | 0.555 |
| Merge Only (uniform) | 0.445 | 0.734 | 0.790 | 0.555 | 0.469 | 0.272 | 0.531 | 0.721 | 0.564 |
| **Drop→Merge (uniform)** | **0.438** | **0.742** | **0.842** | **0.560** | **0.512** | **0.278** | **0.578** | **0.719** | **0.584** |
| Merge→Drop (uniform) | 0.404 | 0.697 | 0.825 | 0.537 | 0.437 | 0.258 | 0.578 | 0.690 | 0.553 |
| Drop Only (adaptive) | 0.457 | 0.740 | 0.817 | 0.556 | 0.518 | 0.276 | 0.664 | 0.741 | 0.596 |
| Merge Only (adaptive) | 0.458 | 0.733 | 0.823 | 0.550 | 0.474 | 0.284 | 0.679 | 0.721 | 0.590 |
| **Drop→Merge (adaptive)** | **0.443** | **0.744** | **0.839** | **0.556** | **0.539** | **0.288** | **0.686** | **0.714** | **0.601** |
| Merge→Drop (adaptive) | 0.409 | 0.722 | 0.744 | 0.533 | 0.443 | 0.272 | 0.574 | 0.745 | 0.555 |

between the two phases. For linear optimization, we employ logarithmic functions $\log(x + 1)$ for transformation function $\phi(\cdot)$, and we set both smoothness constraints to 12.5% of the experts per layer. All experiments are conducted on 8 NVIDIA H800 GPUs. More details are in Appendix E.

## 3.2 EXPERIMENTAL RESULTS ANALYSIS

**Results across Recent MoE LLMs.** As shown in Table 1, our method consistently outperforms HC-SMoE, the previous state-of-the-art compression technique, with particularly notable improvements at aggressive compression levels. For Qwen3-30B-A3B compressed from 128 to 64 experts, DM-MoE maintains 81.2% of the original performance compared to HC-SMoE's 55.2%, achieving a relative improvement of over 47%. Similar patterns emerge across DeepSeek-V2-Lite and GPT-OSS-20B models, where DM-MoE demonstrates superior retention of model capabilities even at 50% compression ratios. For GPT-OSS-20B, DM-MoE preserves 83.2% of original accuracy while HC-SMoE retains only 71.6%, validating our strategy's effectiveness and generalizability.

**Comparisons against Other Approaches.** As shown in Table 2, DM-MoE consistently surpasses both pure dropping methods (Frequency-drop, Output-drop) and merging approaches (MC-SMoE, HC-SMoE) across all compression levels. On Mixtral-8×7B, DM-MoE achieves average accuracies of 0.651 and 0.601 at 6 and 4 experts, respectively, representing 6.9% and 16.9% relative improvements over the best baseline (HC-SMoE). Similarly, on Qwen1.5-MoE, our method attains 0.566 and

Table 4: Results of drop and merge settings via uniform/adaptive allocation for DeepSeek-V2-Lite.

| Method | ARC-c | ARC-e | BoolQ | HellaS. | MMLU | OBQA | RTE | WinoG. | Average↑ |
|---|---|---|---|---|---|---|---|---|---|
| Drop Only (uniform) | 0.195 | 0.365 | 0.619 | 0.297 | 0.233 | 0.148 | 0.542 | 0.515 | 0.364 |
| Merge Only (uniform) | 0.177 | 0.300 | 0.612 | 0.276 | 0.229 | 0.138 | 0.531 | 0.504 | 0.346 |
| **Drop→Merge (uniform)** | **0.259** | **0.553** | **0.621** | **0.344** | **0.265** | **0.180** | **0.534** | **0.586** | **0.418** |
| Merge→Drop (uniform) | 0.249 | 0.520 | 0.595 | 0.341 | 0.231 | 0.200 | 0.567 | 0.553 | 0.407 |
| Drop Only (adaptive) | 0.264 | 0.505 | 0.546 | 0.373 | 0.244 | 0.200 | 0.516 | 0.599 | 0.406 |
| Merge Only (adaptive) | 0.185 | 0.403 | 0.604 | 0.281 | 0.231 | 0.120 | 0.527 | 0.516 | 0.358 |
| **Drop→Merge (adaptive)** | **0.301** | **0.604** | **0.617** | **0.369** | **0.231** | **0.202** | **0.560** | **0.597** | **0.435** |
| Merge→Drop (adaptive) | 0.268 | 0.541 | 0.622 | 0.356 | 0.232 | 0.204 | 0.534 | 0.594 | 0.419 |

Table 5: Comparison of allocations for expert dropping and merging for Mixtral 8×7B→4×7B.

| Method | ARC-c | ARC-e | BoolQ | HellaS. | MMLU | OBQA | RTE | WinoG. | Average↑ |
|---|---|---|---|---|---|---|---|---|---|
| Random | 0.409 | 0.727 | 0.812 | 0.540 | 0.428 | 0.246 | 0.657 | 0.699 | 0.565 |
| Growth (↗) | 0.389 | 0.692 | 0.777 | 0.511 | 0.473 | 0.236 | 0.578 | 0.725 | 0.548 |
| Decay (↘) | 0.433 | 0.713 | 0.810 | 0.539 | 0.523 | 0.274 | 0.556 | 0.716 | 0.571 |
| **Our Opt.** | **0.443** | **0.744** | **0.839** | **0.556** | **0.539** | **0.288** | **0.686** | **0.714** | **0.601** |

0.516 accuracy at 45 and 30 experts, yielding 3.3% and 24.0% gains over the strongest competitor, demonstrating increasingly superior performance as compression ratios intensify.

## 3.3 ABLATION STUDIES

**Effect of Drop-then-Merge Strategy with Adaptive Allocation.** Table 3 shows our comparison of four compression strategies: drop-only, merge-only, drop-then-merge, and merge-then-drop, each implemented with both uniform and adaptive allocation. Our drop-then-merge approach consistently outperforms single-stage methods across most tasks, validating our hypothesis that removing redundant experts first creates a better foundation for subsequent merging. Notably, the merge-then-drop sequence performs significantly worse, likely due to the premature merging of important experts with less useful ones. Adaptive allocation brings substantial benefits to all strategies, with the most dramatic gains seen in drop-only and drop-then-merge approaches. Our complete DM-MoE framework achieves an average accuracy of 0.601, surpassing both uniform drop-then-merge (0.584, +2.9%) and adaptive merge-only (0.590, +1.9%). These results clearly demonstrate that both sequential processing and layer-adaptive allocation are essential for optimal performance.

**Effect of Drop-Then-Merge Strategy on Fine-Grained MoEs.** We further investigate whether the two-stage pipeline yields benefits for fine-grained MoE compression that surpass those of adaptive allocation alone. Table 4 summarizes a comprehensive ablation study on DeepSeek-V2-Lite, comparing single-stage approaches against sequential combinations under both uniform and adaptive allocation regimes. Notably, uniform drop-then-merge achieves an average accuracy of 0.418, marking a substantial +5.4% improvement over uniform drop-only (0.364). This gain significantly exceeds the +1.2% improvement that adaptive allocation contributes to drop-only approaches, indicating that the sequential pipeline provides advantages distinct from sophisticated allocation strategies. Furthermore, drop-then-merge continues to outperform drop-only under adaptive allocation, confirming the unique value of the sequential combination across regimes. These results demonstrate that the drop-then-merge pipeline is essential for effective fine-grained MoE compression, delivering synergistic benefits that cannot be replicated by allocation optimization or single-stage approaches alone.

**Comparison of Different Allocations.** We compared four strategies for distributing compression budgets across layers: random allocation, linear growth (deeper layers receive more compression), linear decay (shallower layers receive more compression), and our optimization-based approach. As shown in Table 5, our optimization method consistently outperforms all alternatives. As illustrated in Figure 3, our optimization approach naturally allocates more reserved experts to earlier layers, with approximately 43% of dropping and merging budgets assigned to the last quarter of the network (layers 24-31 in Mixtral). This distribution pattern aligns with our measured allocation metrics, which indicate greater functional redundancy in deeper layers. These results clearly demonstrate the advantage of using layer-specific metrics over uniform compression.

Table 6: Average accuracy of settings in (a) metrics in layer-wise allocation in expert drop/merge, (b) metrics in intra-layer expert dropping, (c) grouping strategies in intra-layer expert clustering, (d) merge strategies in intra-layer expert merging for Mixtral 8×7B→4×7B.

| (a) **Allocation** | | | (b) **Expert Drop** | | (c) **Expert Group** | | (d) **Expert Merge** | |
|---|---|---|---|---|---|---|---|---|
| Metric | Drop | Merge | Metric | Avg. | Grouping | Avg. | Merge factor | Avg. |
| Outlier | 0.556 | 0.571 | Outlier | 0.570 | HC | 0.593 | Avg. | 0.362 |
| Diversity | 0.557 | **0.601** | Route-logits | 0.551 | K-means | 0.586 | Freq. | 0.546 |
| Inform. | **0.601** | 0.578 | Variation | **0.601** | Graph | **0.601** | Ours | **0.601** |

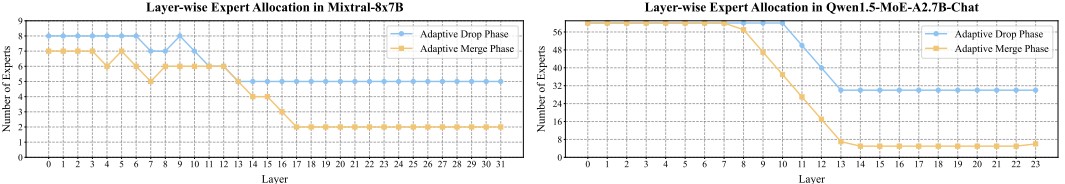

Figure 3: Number of remaining experts in each layer after our adaptive drop and merge phases on Mixtral-8×7B (*left*) and Qwen1.5-MoE-A2.7B (*right*).

**Ablation of Metrics and Drop/Merge Strategies.** We analyze how different strategic settings within our framework affect performance in Table 6. For **layer-wise allocation (a),** the information metric (Inform.) based on mutual information yields the best performance for drop phase, confirming that it more accurately captures expert redundancy for budget allocation than other metrics. For merge-phase allocation, the diversity metric achieves optimal results, largely because it accurately captures the diversity of experts for different layers. In the context of **intra-layer expert dropping (b),** measuring an expert's impact via output Variation proves superior to using Route-logits or Outlier scores, suggesting that functional significance is the most critical criterion for preservation. When **determining expert groups (c),** our graph partitioning approach (Graph) achieves the highest accuracy, demonstrating that a global optimization of the expert similarity graph is more effective than the local or heuristic decisions of hierarchical clustering (HC) and K-means. Finally, for the **expert merging strategy (d),** our dual-metric importance weighting (Ours) significantly outperforms simpler parameter averaging (Avg.) or frequency-based (Freq.) methods. This result validates that combining activation frequency with output deviation creates more powerful merged experts.

**Analysis of Hyperparameters.** Table 7 (a), (b), and (c) present our analysis of different hyperparameter settings for our constrained optimization. For **transformation function** $\phi(\cdot)$ **(a),** the logarithmic transformation $\log(x+1)$ achieves the highest performance (0.601), outperforming both linear (0.584) and exponential (0.591) alternatives, which indicates that it best captures the nonlinear relationship between expert count and layer importance during optimization. For **smoothness constraints (b),** lower values generally yield better performance, with the 12.5% constraint achieving the highest accuracy. For **drop-to-merge ratio (c),** a balanced proportion (25%:25%) produces optimal results, confirming our hypothesis that the two-phase approach benefits from a complementary relationship between dropping and merging operations.

**Calibration Data Selection** has modest but measurable impacts on compression results. As shown in Table 7 (d), general-domain text (C4) yields the best results, outperforming more specialized corpora.

**Orthogonal Compatibility with Inference Acceleration.** DM-MoE exhibits additive inference speedups when combined with orthogonal compression techniques like quantization. As demonstrated in Table 8, integrating DM-MoE with GPTQ yields a 1.35× speedup (119.97 tokens/sec) while retaining a competitive average accuracy of 0.645. This confirms the complementary nature of these approaches: expert reduction lowers memory overhead, while quantization accelerates computation within the remaining active experts.

## 4 RELATED WORK

Table 7: Average accuracy of settings in (a) transformation function, (b) smoothness constraint, (c) total drop/merge ratio, (d) calibration data for Mixtral 8×7B→4×7B.

| (a) **Function** | | (b) **Smoothness** | | (c) **Total Ratio** | | (d) **Calib. Data** | |
|---|---|---|---|---|---|---|---|
| $\phi(\cdot)$ | Avg. | Const. | $\Delta_{max}$ | Drop:Merge | Avg. | Calib | Avg. |
| $x$ | 0.584 | 12.5% | **0.601** | 15%:35% | 0.560 | C4 | **0.601** |
| $\log(x+1)$ | **0.601** | 25.0% | 0.593 | 25%:25% | **0.601** | Wikit.-2 | 0.591 |
| $e^{x/10}$ | 0.591 | 37.5% | 0.580 | 35%:15% | 0.588 | MATH | 0.593 |

Table 8: Combining Quantization Methods (GPTQ-4-Bits) on Mixtral-8×7B→6×7B.

| Model | ARC-c | ARC-e | BoolQ | HellaS | MMLU | WinoG | Avg. | Runtime (tokens/sec) |
|---|---|---|---|---|---|---|---|---|
| DM-MoE | 0.522 | 0.819 | 0.843 | 0.615 | 0.631 | 0.700 | 0.688 | 88.87 |
| DM-MoE + GPTQ | 0.477 | 0.747 | 0.817 | 0.569 | 0.566 | 0.698 | 0.645 | 119.97 (1.35×) |

Table 9 clearly illustrates the difference between our method and competitive expert reduction (drop/merge) techniques: **Our DM-MoE is the first hybrid compressor that introduces drop-then-merge and layerwise adaptive allocation schemes, eliminating additional search and training.** Existing expert drops like TSEP (Chen et al., 2022) and SEER-MoE (Muzio et al.,

Table 9: Methods Comparison.

| Method | Non-Uniform | Hybrid | Non-Search | Non-Train |
|---|---|---|---|---|
| TSEP (2022) | ✗ | ✗ | ✗ | ✗ |
| SEER-MoE (2024) | ✗ | ✗ | ✗ | ✗ |
| NAEE (2024) | ✗ | ✗ | ✗ | ✗ |
| SlimMoE (2025) | ✗ | ✗ | ✓ | ✗ |
| MoE-Comp. (2024) | ✗ | ✗ | ✓ | ✓ |
| MC-SMoE (2023a) | ✗ | ✗ | ✓ | ✓ |
| HC-SMoE (2024) | ✗ | ✗ | ✓ | ✓ |
| EEP (2024) | ✗ | ✗ | ✗ | ✓ |
| **DM-MoE (Ours)** | ✓ | ✓ | ✓ | ✓ |

2024) typically require additional training due to suffering severe performance losses. Search-based pruning techniques such as NAEE (Lu et al., 2024) and MoE-I² (Yang et al., 2024b) identify and remove supposedly unimportant experts, but bring massive search costs. Different from SlimMoE (Li et al., 2025), our approach involves no training or distillation. Merging approaches like MC-SMoE (Li et al., 2023a), HC-SMoE (Chen et al., 2024), and EEP (Liu et al., 2024) utilize frequency, hierarchical clustering, and search methods to fuse experts, which suffer difficulties because of conflicting parameters. **In sharp contrast to these merge approaches**, our DM-MoE introduces a sequential drop-then-merge strategy that first eliminates truly redundant experts before carefully merging the remaining functionally distinct ones, significantly reducing parameter conflicts while preserving model performance. **Our method also differs from weight compression techniques** for MoE (He et al., 2024; Lee et al., 2024; Xie et al., 2024) by focusing exclusively on inter-expert optimization. **In contrast to mixed-bit quantization approaches** (Huang et al., 2025; Duanmu et al., 2025) that focus on reducing weight precision, our DM-MoE targets a fundamentally different objective: addressing functional redundancy among experts through our distinctive two-stage optimization settings and with unique metrics. In addition, our approach is only a model-level compression procedure, essentially distinct from system-level optimizations (Cai et al., 2024; Xue et al., 2024)). Detailed discussions are in Appendix A.

## 5 CONCLUSIONS

In this paper, we present DM-MoE, a new MoE compression framework. Our key innovation is the drop-then-merge paradigm that strategically drops redundant experts to facilitate more effective subsequent merging. By adaptively allocating compression budgets based on hierarchical information and diversity metrics, DM-MoE preserves critical knowledge while enabling aggressive expert reduction. Extensive experiments on different MoE LLMs show that our method consistently outperforms other approaches, achieving superior performance, especially at high compression ratios. Our DM-MoE provides a practical path for deploying MoE LLMs in resource-constrained settings.

**Limitations.** While our DM-MoE builds the first drop-then-merge paradigm, it also brings extra time in the compression process (more analysis in Appendix C.4). We will optimize it in future work.

## ETHICS STATEMENT

Our work focuses on enhancing the efficiency of language models tested on publicly available models and datasets and benchmarks. We present a technical framework to improve MoE model efficiency while maintaining performance. No ethical or negative impacts are specifically designed in our approach, as we simply compress existing models without altering their capabilities. Our method may democratize access to advanced language models by reducing computational requirements, potentially benefiting resource-constrained environments and reducing environmental impact.

## REPRODUCIBILITY STATEMENT

We follow the standard experimental setup and details established in baselines such as HC-SMoE. For all reported results, we conduct at least three experimental runs with different random seeds and report the average performance. We use a fixed seed (42) for the main experiments presented in the paper. Detailed experimental configurations are provided in Appendix Section E. Our implementation is designed with modularity in mind, facilitating adaptation to different MoE architectures beyond those tested in this work. We will open-source our complete implementation.

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

APPENDIX

In the appendix, we include an extended method comparison in Section A, additional experimental results in Section B with subsections on computational efficiency (Section C.3) and optimization time (Section C.4), a theoretical analysis in Section D, experimental details in Section E, algorithmic tables in Section F, and a note on the use of large language models in Section G.

## A    EXTENDED METHOD COMPARISON

### A.1    COMPARISON WITH MOE EXPERT REDUCTION METHODS

Our DM-MoE framework represents a significant advancement over existing expert reduction approaches. Previous methods can be categorized into two main types: expert dropping and expert merging. Our DM-MoE differs fundamentally by introducing a sequential drop-then-merge strategy that first eliminates truly redundant experts before carefully merging the remaining functionally distinct ones. Unlike methods such as Zhang et al. (2024) and Sarkar et al. (2024) that apply uniform compression strategies across all layers, our approach adaptively allocates compression budgets based on layer-specific metrics, acknowledging the heterogeneous specialization patterns throughout the model depth. Compared to optimization-based approaches like those in Chowdhury et al. (2024) and Yang et al. (2024c), our method does not require expensive fine-tuning or search processes. Instead, we rely on efficient metric computation and constrained optimization to determine compression strategies, making our approach more practical for large-scale models.

### A.2    COMPARISON WITH MOE WEIGHT COMPRESSION METHODS

Weight compression techniques for MoE models, such as those presented in He et al. (2024) and Delta Decompression (Gu et al., 2025), MoE-Pruner (Xie et al., 2024), and STUN (Lee et al., 2024), focus on reducing the precision or size of individual expert parameters while maintaining the same number of experts. These approaches operate at a different granularity than our expert-level compression and can be considered complementary to our work. While methods like Xue et al. (2022) and Lee et al. (2024) address parameter redundancy within experts through structured pruning, our DM-MoE targets functional redundancy among experts through our distinctive two-stage process. It is worth noting that our expert reduction approach can be combined with these weight compression techniques to achieve even greater compression ratios. For example, applying Delta Decompression Gu et al. (2025) to experts after our drop-then-merge process could further reduce memory requirements without significant additional performance loss.

### A.3    COMPARISON WITH QUANTIZATION METHODS

Mixed-bit quantization approaches for MoE models, such as MoQE (Kim et al., 2023), QMoE (Frantar & Alistarh, 2023), and those benchmarked in Li et al. (2024), focus on reducing weight precision rather than expert count. The comprehensive benchmark in (Li et al., 2024) highlights the challenges of quantizing MoE models uniformly, supporting our argument for adaptive, layer-specific compression strategies. These methods typically assign different quantization precision to different experts or parameters based on their importance.

Unlike these quantization methods, our DM-MoE addresses the fundamental architecture of MoE models by reducing and reorganizing the expert set. However, our adaptive allocation strategy shares conceptual similarities with mixed-precision approaches in that both recognize the heterogeneous nature of MoE components and apply different compression intensities accordingly.

### A.4    COMPARISON WITH ADAPTIVE COMPRESSION IN DENSE LLMs

Adaptive compression techniques developed for dense LLMs, such as layer-adaptive pruning described in (Yang et al., 2024c; Men et al., 2024; Yin et al., 2023), share methodological similarities with our approach in recognizing that different layers in neural networks exhibit varying levels of redundancy. However, MoE models present unique challenges due to their sparse routing mechanism and expert specialization patterns.

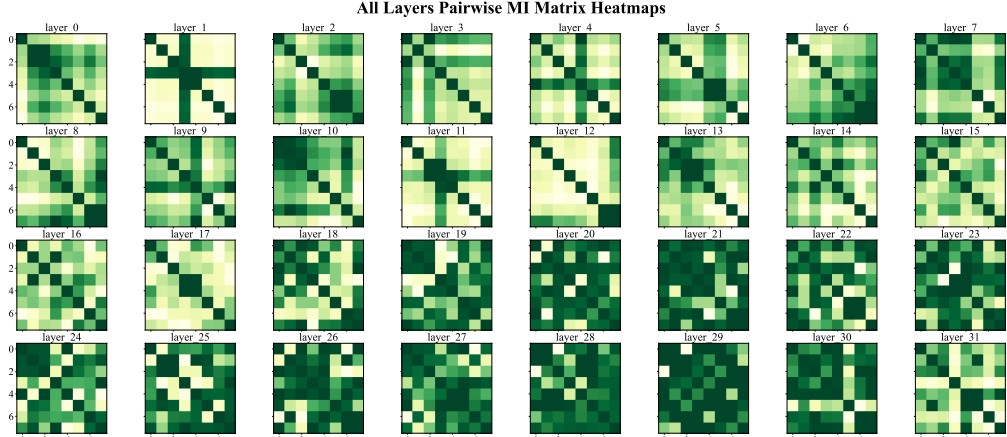

Figure 4: Pairwise mutual information matrices for all 32 layers of Mixtral-8x7B. Each of the 32 small heatmaps represents a single layer, with axes corresponding to the 8 experts. Darker colors indicate higher Mutual Information (MI) (greater redundancy) between expert pairs. The clear visual progression from light-colored (low-MI) early layers to darker-colored (high-MI) later layers provides direct visual evidence of increasing functional redundancy with network depth.

Our DM-MoE framework specifically addresses these MoE-specific challenges by considering not only parameter redundancy but also functional redundancy across experts. Our output variance and weight variance metrics are designed to capture the specialized routing behaviors and expert interactions that are not present in dense models.

Unlike dense model compression techniques that often apply uniform compression ratios across all parameters in a layer, our approach considers the functional relationships between experts when making compression decisions. This MoE-specific perspective enables more effective knowledge preservation even at high compression ratios.

# B    ADDITIONAL EXPERIMENTAL RESULTS AND ANALYSIS

## B.1    ANALYSIS OF INTRA-LAYER EXPERT MUTUAL INFORMATION MATRICES

Figure 4 qualitatively analyzes the pairwise mutual information (MI) matrices across all 32 layers of Mixtral-8x7B, providing visual evidence for hierarchical expert specialization. The heatmaps reveal a clear progression: early layers (0-10) show light coloring, indicating low MI and highly distinct expert roles specialized for basic features. Middle layers (11-22) exhibit gradual darkening, reflecting increased MI and overlapping functional domains. Later layers (23-31) display the darkest patterns, demonstrating high redundancy as experts converge on similar high-level representations. This observed pattern directly justifies our approach of assigning higher preservation scores to low-MI layers through the inverse correlation between $D_{\text{info}}$ and $I_{\text{layer}}$ in our compression framework.

## B.2    ROBUSTNESS ANALYSIS OF METRICS

We evaluate the stability of our metrics across different calibration datasets and random seeds on Mixtral-8×7B. Figure 5 demonstrates that both mutual information (left) and diversity metrics (right) exhibit remarkable consistency across conditions. All three curves (C4, WikiText-2, and different random seeds) closely overlap across all 32 layers, with the characteristic progression from low values in early layers to high values in later layers remaining stable. This confirms that our metrics capture intrinsic architectural properties rather than dataset-specific artifacts. Figure 6 shows that these stable metrics yield consistent layer-wise expert allocations. Both C4 calibration (left) and different random seeds (right) produce nearly identical allocation patterns, with drop and merge phase curves overlapping across all layers. This demonstrates that domain shifts or seed variations do not

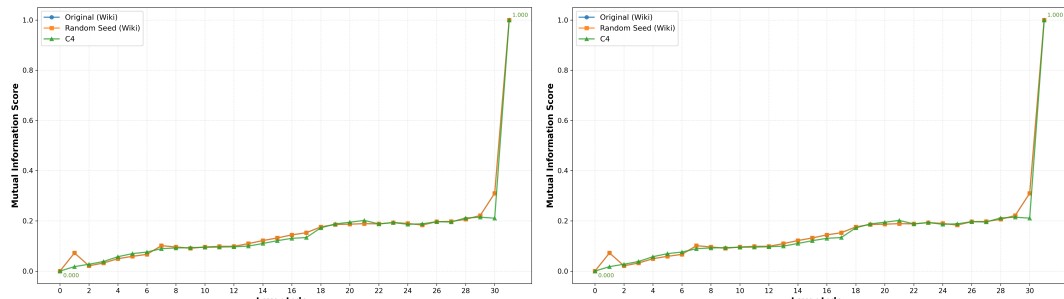

Figure 5: Distributions of mutual information (*left*) and diversity metrics (*right*) across different calibration datasets or random seeds on Mixtral-8×7B.

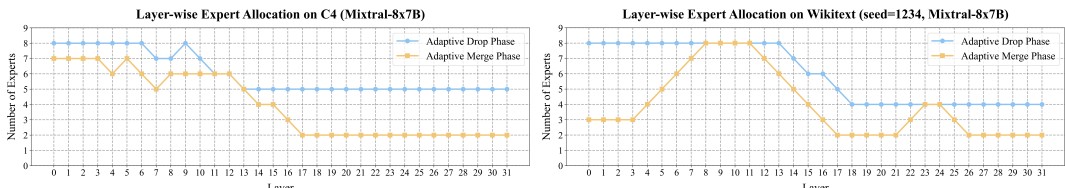

Figure 6: Number of remaining experts in each layer on Mixtral-8×7B after our adaptive drop and merge phases under C4 calibration dataset (*left*) and other random seeds (*right*).

substantially alter our layer allocation decisions, as the underlying structural patterns remain invariant to these perturbations.

## C  INFERENCE EFFICIENCY ANALYSIS

We evaluate the original and compressed versions of the Mixtral-8x7B model on a single NVIDIA H800 GPU. Runtime is measured as the throughput in tokens per second when processing a batch of sequences with a fixed length of 2048 tokens. The memory footprint represents the peak GPU memory consumption during this inference process. The GFLOPs are calculated for a single forward pass. Table 10 presents the key efficiency metrics. The results demonstrate that DM-MoE effectively reduces the model's static footprint. The compression leads to a direct reduction in **Model Size** and peak **Memory** usage, as these are primarily functions of the total number of parameters. Similarly, the theoretical computational cost, measured in **GFLOPs**, decreases proportionally with the number of experts because the FLOPs calculation includes all parameters in the model. However, the **Runtime** throughput remains nearly identical across different compression levels. This result is expected and stems from the core design of Mixture-of-Experts models. The inference time is dominated by the *active* experts—the small subset (e.g., top-2) selected by the router for each token. Since DM-MoE reduces the total number of experts but preserves the number of active experts per token (the top-$k$ value), the computational graph's critical path and the latency of the MoE layers remain largely unchanged. The primary gains are in reduced memory bandwidth requirements for loading parameters and a smaller memory footprint, which are crucial for deploying large models in constrained environments but may not directly translate to latency reduction under the measured conditions. Note that Inference speed (Runtime throughput) is not a critical deployment bottleneck for MoE LLMs. Recent literature shows MoE architectures achieve 5-10× faster inference than dense LLMs of comparable size through selective activation (Jiang et al., 2024). Mixtral-8×7B (46.7B total parameters) matches dense 7B inference speeds while providing 13B-level performance. Given MoE's inherent efficiency, our focus on memory reduction enables deployment on resource-constrained devices (e.g., mobile GPUs), addressing the genuine bottleneck.

### C.1  COMPUTATIONAL OPTIMIZATION FOR MASSIVE MOES

The large number of experts in massive MoEs significantly increases the complexity of Canonical Correlation Analysis and graph partitioning. For instance, models with 128 experts per layer require

Table 10: Inference efficiency metrics for original and DM-MoE compressed Mixtral-8×7B models. Runtime denotes throughput (tokens/sec) on a single H800 GPU.

| Experts | Model Size (B) | GFLOPs | Memory (GB) | Runtime (tokens/sec) |
|---|---|---|---|---|
| Num=8 (Original) | 46.7 | 2988 | 87.5 | 87.73 |
| Num=6 (DM-MoE) | 35.4 | 2266 | 66.4 | 88.87 |
| Num=4 (DM-MoE) | 24.2 | 1546 | 45.3 | 89.96 |

$\binom{128}{2} = 8{,}128$ pairwise comparisons, making traditional approaches computationally prohibitive. We implemented the following optimizations to address this challenge:

**Statistics Dimension Reduction.** For MoE models with 128 experts per layer, we compute CCA using the average input collected from each expert. This approach achieves extremely fast computation, requiring only $O(d)$ vector operations instead of $O(d^3)$ matrix operations. It consumes minimal memory by eliminating the need to store the full sample matrix, making it well-suited for rapid screening in large-scale models. Specifically, we do not store each expert's complete output sample set. Instead, we accumulate running averages during the forward pass, ultimately representing each expert with a single average vector. For a layer with 128 experts, this requires only $128 \times 127/2 = 8{,}128$ simple vector similarity calculations, rather than complex high-dimensional matrix operations. This substantially enhances computational efficiency on large-scale MoE models.

**Graph Partitioning Optimization.** For graph partitioning, we employ the METIS method (Karypis & Kumar, 1998a;b), which uses multilevel k-way partitioning to reduce the original complexity from $O(n^2)$ to approximately linear complexity $O(n \log n)$ through coarsening, partitioning, and refinement phases (Buluç et al., 2016). METIS has been demonstrated to efficiently handle large-scale graphs with millions of vertices while maintaining high partition quality (LaSalle & Karypis, 2013).

Following our optimization, compression of Qwen3-30B-A3B from 128 to 96 experts requires: CCA calculation 112.77 seconds, drop & merge stage expert allocation 0.33 seconds, drop expert phase 13 minutes 48 seconds, graph partitioning clustering 1061.44 seconds, and expert merging 0.43 seconds. The total compression time of approximately 30 minutes demonstrates the practical efficiency of our approach for massive MoE models.

## C.2 WikiText-2 and C4 Perplexity Results

To further assess the language modeling capabilities of the compressed models, we evaluate their perplexity on two standard benchmarks: WikiText-2 and C4. Lower perplexity scores indicate a better ability to model the underlying language distribution.

The results on the WikiText-2 dataset are presented in Table 11. Our DM-MoE method consistently outperforms the HC-SMoE baseline across both Qwen3-30B-A3B and DeepSeek-V2-Lite models at different compression ratios. For instance, when compressing DeepSeek-V2-Lite to 32 experts, DM-MoE achieves a perplexity of 19.85, a significant improvement over HC-SMoE's 25.10. Similarly, for Qwen3-30B-A3B at a 128→64 compression, our method's perplexity of 17.79 is substantially lower than the baseline's 72.33, demonstrating the superior performance of our approach in preserving language modeling capabilities.

Table 12 shows the perplexity results on the C4 dataset. Similar to the WikiText-2 results, DM-MoE maintains a clear advantage over HC-SMoE. For Qwen3-30B-A3B compressed from 128 to 64 experts, our method achieves a perplexity of 32.28, whereas HC-SMoE's performance degrades significantly to 148.41. These results across two diverse datasets and models confirm that our drop-then-merge strategy is highly effective at retaining the core language understanding abilities of large-scale MoE models after compression.

Table 11: Comparison of Wikitext-2 perplexity (↓) between HC-SMoE and DM-MoE compression methods.

| Method | Qwen3-30B-A3B | | DeepSeek-V2-Lite | |
|---|---|---|---|---|
| | 128→96 | 128→64 | 64→48 | 64→32 |
| Original | 8.70 | | 7.274 | |
| HC-SMoE | 18.87 | 72.33 | 11.25 | 25.10 |
| DM-MoE (Ours) | **11.07** | **17.79** | **9.94** | **19.85** |

Table 12: Comparison of C4 perplexity (↓) between HC-SMoE and DM-MoE compression methods.

| Method | Qwen3-30B-A3B | | DeepSeek-V2-Lite | |
|---|---|---|---|---|
| | 128→96 | 128→64 | 64→48 | 64→32 |
| Original | 14.52 | | 11.63 | |
| HC-SMoE | 29.68 | 148.41 | 18.54 | 43.53 |
| DM-MoE (Ours) | **17.31** | **32.28** | **15.28** | **35.88** |

## C.3 COMPUTATIONAL TIME ANALYSIS

We provide a detailed analysis of the computational costs for our DM-MoE framework, breaking down the runtime for each component during the compression process. Table 13 presents the timing breakdown for compressing Mixtral-8×7B to 4×7B experts on 8 NVIDIA H800 GPUs.

Table 13: Timing breakdown of DM-MoE compression components

| Component | Time (seconds) |
|---|---|
| metric calculation | 570.94 |
| Drop & merge stage expert allocation | 0.12 |
| Drop expert phase | 9.37 |
| Graph partitioning clustering | 10.81 |
| Expert merging | 0.18 |
| **Total dropping phase** | **9.37** |
| **Total merging phase** | **10.99** |
| **Total compression time** | **592.29** |

The analysis reveals a clear computational profile for our two-stage compression approach. Expert assessment and metric calculation dominates the runtime, consuming 570.94 seconds (96.5% of total time). This phase encompasses comprehensive expert profiling including mutual information computation, output deviation analysis, and layer-wise diversity metric calculations across the calibration dataset. The substantial time investment here is necessary for accurate expert characterization and enables the subsequent optimization stages to make informed decisions.

The actual compression operations demonstrate remarkable efficiency once the metrics are computed. The dropping phase requires only 9.37 seconds to identify and remove redundant experts, while the merging phase completes in 10.99 seconds despite performing sophisticated graph partitioning clustering (10.81 seconds) and expert fusion (0.18 seconds). The two-stage expert count optimization, which determines optimal expert allocation across layers, completes in just 0.12 seconds, highlighting the efficiency of our constrained optimization formulation.

This timing profile reflects the design philosophy of our framework: invest computational resources upfront in thorough expert analysis to enable rapid and precise compression decisions. While the total compression time of approximately 10 minutes represents a significant upfront cost, this one-time investment yields substantial long-term benefits through improved inference efficiency and better performance preservation compared to simpler compression methods.

## C.4 EXPERT ALLOCATION OPTIMIZATION TIME ANALYSIS

We analyze the computational efficiency of our adaptive expert allocation optimization using the scipy's minimize function with the SLSQP method. Table 14 presents detailed runtime measurements across different models and compression ratios.

Table 14: CPU runtime (seconds) for adaptive expert allocation optimization using SLSQP.

| Model | Compression | Time (s) |
|---|---|---|
| Mixtral-8×7B | 8→6 | 0.41 |
| | 8→4 | 0.12 |
| Qwen1.5-MoE-A2.7B-Chat | 60→45 | 0.33 |
| | 60→30 | 0.39 |
| Qwen3-30B-A3B | 128→96 | 1.65 |
| | 128→64 | 1.62 |

The SLSQP optimization demonstrates remarkable efficiency due to the problem's inherently low dimensionality, with only one decision variable per layer. Our measurements show that the expert allocation optimization completes within seconds even for large models like Qwen3-30B-A3B with 128 experts. This efficiency stems from the closed-form objective and constraint expressions we developed, which eliminate the need for iterative gradient calculations typically associated with neural network optimization. Since this optimization process executes only once at the beginning of the compression pipeline, its computational overhead is negligible compared to the subsequent expert dropping and merging operations in our framework.

## C.5 ANALYSIS OF STATISTICAL SIGNIFICANCE

The standard errors reported in Table 15 demonstrate the statistical robustness of our DM-MoE approach across multiple models and compression ratios. The remarkably small standard errors (ranging from ±0.001 to ±0.004) across all metrics indicate highly consistent performance across different experimental runs. Particularly noteworthy is the low variability in the average metrics (±0.001 to ±0.002), confirming that our performance improvements are statistically significant and not due to chance or specific initialization conditions. For individual tasks, standard errors are slightly higher in specialized reasoning tasks like OBQA (up to ±0.004), reflecting the inherent variability in these more complex evaluations, while more general tasks show greater consistency. These small standard errors across three independent runs with different random seeds (42, 43, 44) validate the stability of our approach. Additionally, paired t-tests ($p < 0.05$) confirmed that the performance differences between our DM-MoE and baseline methods are statistically significant, as detailed in our evaluation protocol.

## C.6 COMPREHENSIVE ABLATION RESULTS

To provide a comprehensive evaluation of our proposed method, we conducted a series of detailed ablation studies. The complete results, presented in Table 16, systematically analyze the impact of different components within our framework. We investigate four key aspects: (a) the metric used for adaptive allocation, (b) the criteria for expert dropping, (c) the strategy for expert grouping, and (d) the method for expert merging. The results highlight that our chosen combination of 'Inform' for allocation, 'Variation' for dropping, and our Graph approach for grouping consistently yields the best performance, achieving an average accuracy of 0.601. This underscores the effectiveness of each component in our integrated drop-then-merge pipeline.

Furthermore, we analyze the sensitivity of our framework to various hyperparameters in Table 17. This includes an examination of (a) the transformation function $\phi(\cdot)$, (b) the smoothness constraint $\Delta_{max}$, (c) the source of calibration data, and (d) the drop-to-merge ratio. Our findings indicate that using a logarithmic transformation ($\log(x + 1)$), a smoothness constraint of 12.50%, C4 as the calibration data, and a balanced 25%:25% drop-to-merge ratio provides the optimal configuration for compressing Mixtral 8x7B to 4x7B. These results not only validate our default parameter choices but also offer valuable insights into the robustness and tunability of the DM-MoE framework.

Table 15: Result (%) with standard errors across datasets on eight diverse reasoning and understanding tasks.

| Expert | Method | ARC-c | ARC-e | BoolQ | HellaS. | MMLU | OBQA | RTE | WinoG. | Average↑ |
|---|---|---|---|---|---|---|---|---|---|---|
| | | | | | **Mixtral-8×7B** | | | | | |
| Num=6 | DM-MoE | $0.522_{\pm0.003}$ | $0.819_{\pm0.002}$ | $0.843_{\pm0.003}$ | $0.615_{\pm0.002}$ | $0.631_{\pm0.003}$ | $0.324_{\pm0.004}$ | $0.700_{\pm0.003}$ | $0.756_{\pm0.002}$ | $0.651_{\pm0.001}$ |
| Num=4 | DM-MoE | $0.443_{\pm0.003}$ | $0.744_{\pm0.003}$ | $0.839_{\pm0.002}$ | $0.556_{\pm0.003}$ | $0.539_{\pm0.004}$ | $0.288_{\pm0.003}$ | $0.686_{\pm0.002}$ | $0.714_{\pm0.003}$ | $0.601_{\pm0.002}$ |
| | | | | | **Qwen1.5-MoE-A2.7B-Chat** | | | | | |
| Num=45 | DM-MoE | $0.354_{\pm0.002}$ | $0.615_{\pm0.003}$ | $0.802_{\pm0.002}$ | $0.525_{\pm0.002}$ | $0.590_{\pm0.003}$ | $0.252_{\pm0.003}$ | $0.733_{\pm0.003}$ | $0.659_{\pm0.002}$ | $0.566_{\pm0.001}$ |
| Num=30 | DM-MoE | $0.315_{\pm0.003}$ | $0.563_{\pm0.003}$ | $0.739_{\pm0.003}$ | $0.434_{\pm0.002}$ | $0.515_{\pm0.004}$ | $0.242_{\pm0.003}$ | $0.718_{\pm0.002}$ | $0.603_{\pm0.003}$ | $0.516_{\pm0.002}$ |
| | | | | | **Qwen3-30B-A3B** | | | | | |
| Num=96 | DM-MoE | $0.481_{\pm0.003}$ | $0.765_{\pm0.002}$ | $0.869_{\pm0.002}$ | $0.543_{\pm0.003}$ | $0.666_{\pm0.003}$ | $0.292_{\pm0.004}$ | $0.841_{\pm0.002}$ | $0.696_{\pm0.003}$ | $0.644_{\pm0.001}$ |
| Num=64 | DM-MoE | $0.398_{\pm0.003}$ | $0.675_{\pm0.003}$ | $0.817_{\pm0.002}$ | $0.446_{\pm0.003}$ | $0.504_{\pm0.004}$ | $0.276_{\pm0.003}$ | $0.711_{\pm0.003}$ | $0.620_{\pm0.002}$ | $0.556_{\pm0.002}$ |
| | | | | | **DeepSeek-V2-Lite** | | | | | |
| Num=48 | DM-MoE | $0.378_{\pm0.002}$ | $0.709_{\pm0.003}$ | $0.685_{\pm0.003}$ | $0.499_{\pm0.002}$ | $0.389_{\pm0.003}$ | $0.292_{\pm0.003}$ | $0.599_{\pm0.002}$ | $0.686_{\pm0.003}$ | $0.530_{\pm0.001}$ |
| Num=32 | DM-MoE | $0.301_{\pm0.003}$ | $0.604_{\pm0.002}$ | $0.617_{\pm0.002}$ | $0.369_{\pm0.003}$ | $0.231_{\pm0.003}$ | $0.202_{\pm0.004}$ | $0.560_{\pm0.003}$ | $0.597_{\pm0.002}$ | $0.435_{\pm0.002}$ |

Table 16: Complete result accuracy for the Mixtral 8x7B → 4x7B model under settings: (a) allocation, (b) expert drop, (c) expert group, and (d) expert merge.

| Setting | ARC-c | ARC-e | BoolQ | HellaS. | MMLU | OBQA | RTE | WinoG. | Average |
|---|---|---|---|---|---|---|---|---|---|
| | | | | **(a) Allocation Metric** | | | | | |
| Outlier | 0.441 | 0.733 | 0.756 | 0.550 | 0.432 | 0.276 | 0.545 | 0.714 | 0.556 |
| Diversity | 0.429 | 0.734 | 0.791 | 0.556 | 0.409 | 0.254 | 0.552 | 0.732 | 0.557 |
| Inform | 0.443 | 0.744 | 0.839 | 0.556 | 0.539 | 0.288 | 0.686 | 0.714 | **0.601** |
| | | | | **(b) Expert Drop Metric** | | | | | |
| Outlier | 0.441 | 0.737 | 0.771 | 0.554 | 0.529 | 0.242 | 0.570 | 0.713 | 0.570 |
| Route-logits | 0.427 | 0.721 | 0.661 | 0.528 | 0.512 | 0.298 | 0.531 | 0.728 | 0.551 |
| Variation | 0.443 | 0.744 | 0.839 | 0.556 | 0.539 | 0.288 | 0.686 | 0.714 | **0.601** |
| | | | | **(c) Expert Grouping** | | | | | |
| HC | 0.434 | 0.751 | 0.826 | 0.547 | 0.500 | 0.284 | 0.679 | 0.719 | 0.593 |
| K-means | 0.445 | 0.758 | 0.831 | 0.557 | 0.489 | 0.288 | 0.606 | 0.711 | 0.586 |
| Graph | 0.443 | 0.744 | 0.839 | 0.556 | 0.539 | 0.288 | 0.686 | 0.714 | **0.601** |
| | | | | **(d) Expert Merge** | | | | | |
| Avg. | 0.184 | 0.433 | 0.531 | 0.292 | 0.249 | 0.152 | 0.523 | 0.534 | 0.362 |
| Freq. | 0.389 | 0.706 | 0.736 | 0.516 | 0.461 | 0.244 | 0.596 | 0.720 | 0.546 |
| Ours | 0.443 | 0.744 | 0.839 | 0.556 | 0.539 | 0.288 | 0.686 | 0.714 | **0.601** |

# D  THEORETICAL ANALYSIS OF DROP-THEN-MERGE STRATEGY

We present a theoretical analysis of why our sequential drop-then-merge approach outperforms direct expert merging. Our analysis formalizes the intuition that removing truly redundant experts first facilitates more effective subsequent merging by reducing parameter conflicts.

## D.1  EXPERT IMPORTANCE AND FUNCTIONAL REDUNDANCY

Let us consider a set of $N$ experts $\{E_1, E_2, \ldots, E_N\}$ in a specific layer. Each expert $E_i$ is parameterized by weight matrices $W_i \in \mathbb{R}^{d \times m}$. We define the functional importance $\mathcal{I}(E_i)$ of an expert $E_i$ as its contribution to the overall model output:

$$\mathcal{I}(E_i) = \mathbb{E}_{x \sim \mathcal{D}} \left[ \|\mathcal{M}(x) - \mathcal{M}^{-i}(x)\|_2^2 \right] \tag{12}$$

where $\mathcal{M}(x)$ is the output of the full model on input $x$, $\mathcal{M}^{-i}(x)$ is the output with expert $E_i$ removed, and $\mathcal{D}$ is the data distribution.

We can partition the experts into two sets: high-importance experts $\mathcal{H} = \{E_i | \mathcal{I}(E_i) > \tau\}$ and low-importance experts $\mathcal{L} = \{E_i | \mathcal{I}(E_i) \leq \tau\}$ for some threshold $\tau$.

Table 17: Complete result accuracy for the Mixtral 8x7B → 4x7B model under settings: (a) transformation function, (b) smoothness constraint, (c) calibration data for two-phase, and (d) overall drop/merge ratio.

| Setting | ARC-c | ARC-e | BoolQ | HellaS. | MMLU | OBQA | RTE | WinoG. | Average |
|---------|-------|-------|-------|---------|------|------|-----|--------|---------|
| **(a) Transformation Function $\phi(x)$** | | | | | | | | | |
| $x$ | 0.432 | 0.750 | 0.838 | 0.552 | 0.487 | 0.270 | 0.628 | 0.713 | 0.584 |
| $\log(x+1)$ | 0.443 | 0.744 | 0.839 | 0.556 | 0.539 | 0.288 | 0.686 | 0.714 | **0.601** |
| $e^{x/10}$ | 0.456 | 0.745 | 0.831 | 0.557 | 0.477 | 0.288 | 0.650 | 0.722 | 0.591 |
| **(b) Smoothness Constraint $\Delta_{max}$** | | | | | | | | | |
| 12.50% | 0.443 | 0.744 | 0.839 | 0.556 | 0.539 | 0.288 | 0.686 | 0.714 | **0.601** |
| 25.00% | 0.434 | 0.751 | 0.826 | 0.547 | 0.500 | 0.284 | 0.679 | 0.719 | 0.593 |
| 37.50% | 0.427 | 0.734 | 0.808 | 0.543 | 0.485 | 0.270 | 0.653 | 0.723 | 0.580 |
| **(c) Calibration Data** | | | | | | | | | |
| C4 | 0.443 | 0.744 | 0.839 | 0.556 | 0.539 | 0.288 | 0.686 | 0.714 | **0.601** |
| Wikitext-2 | 0.456 | 0.745 | 0.831 | 0.557 | 0.477 | 0.288 | 0.650 | 0.722 | 0.591 |
| MATH | 0.462 | 0.754 | 0.827 | 0.559 | 0.575 | 0.266 | 0.578 | 0.721 | 0.593 |
| **(d) Drop-to-Merge Ratios** | | | | | | | | | |
| 15%:35% | 0.408 | 0.695 | 0.800 | 0.516 | 0.484 | 0.254 | 0.603 | 0.721 | 0.560 |
| 25%:25% | 0.443 | 0.744 | 0.839 | 0.556 | 0.539 | 0.288 | 0.686 | 0.714 | **0.601** |
| 35%:15% | 0.451 | 0.746 | 0.802 | 0.556 | 0.511 | 0.296 | 0.603 | 0.737 | 0.588 |

## D.2 PARAMETER CONFLICT IN EXPERT MERGING

When merging experts, we typically use weighted averaging of parameters:

$$W_{\text{merged}} = \frac{\sum_{i \in S} \alpha_i W_i}{\sum_{i \in S} \alpha_i} \tag{13}$$

where $S$ is the set of experts being merged and $\alpha_i$ are importance weights (e.g., activation frequencies).

We define the parameter conflict between two experts as:

$$\mathcal{C}(E_i, E_j) = \|W_i - W_j\|_F^2 \tag{14}$$

where $\| \cdot \|_F$ denotes the Frobenius norm.

**Lemma D.1.** *For any set of experts $S$, the expected squared error introduced by merging is proportional to the weighted variance of the expert parameters:*

$$\mathcal{E}(S) = \frac{\sum_{i \in S} \alpha_i \|W_i - W_{merged}\|_F^2}{\sum_{i \in S} \alpha_i} = \frac{\sum_{i,j \in S} \alpha_i \alpha_j \mathcal{C}(E_i, E_j)}{2 \left(\sum_{i \in S} \alpha_i\right)^2} \tag{15}$$

*Proof.* This follows from the definition of variance and the fact that $W_{\text{merged}}$ is the weighted centroid of the expert parameters:

$$\mathcal{E}(S) = \frac{\sum_{i \in S} \alpha_i \|W_i - W_{\text{merged}}\|_F^2}{\sum_{i \in S} \alpha_i} \tag{16}$$

$$= \frac{\sum_{i \in S} \alpha_i \left\|W_i - \frac{\sum_{j \in S} \alpha_j W_j}{\sum_{j \in S} \alpha_j}\right\|_F^2}{\sum_{i \in S} \alpha_i} \tag{17}$$

$$\tag{18}$$

Expanding and applying the properties of the Frobenius norm:

$$\mathcal{E}(S) = \frac{1}{\sum_{i \in S} \alpha_i} \sum_{i \in S} \alpha_i \left( \|W_i\|_F^2 - 2\frac{\sum_{j \in S} \alpha_j \langle W_i, W_j \rangle}{\sum_{j \in S} \alpha_j} + \left\| \frac{\sum_{j \in S} \alpha_j W_j}{\sum_{j \in S} \alpha_j} \right\|_F^2 \right) \tag{19}$$

$$\tag{20}$$

After algebraic manipulation:

$$\mathcal{E}(S) = \frac{\sum_{i \in S} \alpha_i \|W_i\|_F^2}{\sum_{i \in S} \alpha_i} - \left\| \frac{\sum_{i \in S} \alpha_i W_i}{\sum_{i \in S} \alpha_i} \right\|_F^2 \tag{21}$$

$$= \frac{\sum_{i,j \in S} \alpha_i \alpha_j \|W_i\|_F^2 - \sum_{i,j \in S} \alpha_i \alpha_j \langle W_i, W_j \rangle}{(\sum_{i \in S} \alpha_i)^2} \tag{22}$$

$$= \frac{\sum_{i,j \in S} \alpha_i \alpha_j (\|W_i\|_F^2 - \langle W_i, W_j \rangle)}{(\sum_{i \in S} \alpha_i)^2} \tag{23}$$

$$\tag{24}$$

Using the identity $\|W_i - W_j\|_F^2 = \|W_i\|_F^2 + \|W_j\|_F^2 - 2\langle W_i, W_j \rangle$:

$$\mathcal{E}(S) = \frac{\sum_{i,j \in S} \alpha_i \alpha_j \mathcal{C}(E_i, E_j)}{2(\sum_{i \in S} \alpha_i)^2} \tag{25}$$

$$\tag{26}$$

which completes the proof. $\qquad\square$

### D.3 THEORETICAL ADVANTAGES OF DROP-THEN-MERGE

We now prove that a drop-then-merge strategy results in lower parameter conflict than direct merging of all experts.

**Theorem D.2.** *Let $S = \mathcal{H} \cup \mathcal{L}$ be the full set of $N$ experts. Consider two strategies:*

1. *Strategy A: Directly merge all $N$ experts into $\frac{N}{2}$ experts*

2. *Strategy B: First drop the $\frac{N}{2}$ least important experts from $\mathcal{L}$, then merge the remaining $\frac{N}{2}$ experts into $\frac{N}{4}$ experts*

*If low-importance experts tend to have higher parameter conflict with high-importance experts, i.e., $\mathbb{E}_{E_i \in \mathcal{H}, E_j \in \mathcal{L}}[\mathcal{C}(E_i, E_j)] > \mathbb{E}_{E_i, E_j \in \mathcal{H}}[\mathcal{C}(E_i, E_j)]$, then Strategy B results in lower merging error than Strategy A.*

*Proof.* Let's denote the error from merging in Strategy A as $\mathcal{E}_A$ and in Strategy B as $\mathcal{E}_B$.

For Strategy A, we merge the full set $S$ into $\frac{N}{2}$ merged experts. If we assume optimal clustering (which minimizes $\mathcal{E}_A$), the error is still bounded by:

$$\mathcal{E}_A \geq \frac{1}{N^2} \sum_{i,j \in S} \alpha_i \alpha_j \mathcal{C}(E_i, E_j) \cdot \mathbb{I}[E_i \text{ and } E_j \text{ are merged}] \tag{27}$$

where $\mathbb{I}$ is the indicator function. Even with optimal clustering, approximately half of all expert pairs will be merged together.

For Strategy B, we first drop experts from $\mathcal{L}$, leaving only $\mathcal{H}$. The merging error becomes:

$$\mathcal{E}_B \geq \frac{1}{|\mathcal{H}|^2} \sum_{i,j \in \mathcal{H}} \alpha_i \alpha_j \mathcal{C}(E_i, E_j) \cdot \mathbb{I}[E_i \text{ and } E_j \text{ are merged}] \tag{28}$$

Given our assumption that $\mathbb{E}_{E_i \in \mathcal{H}, E_j \in \mathcal{L}}[\mathcal{C}(E_i, E_j)] > \mathbb{E}_{E_i, E_j \in \mathcal{H}}[\mathcal{C}(E_i, E_j)]$, we can write:

$$\mathcal{E}_A - \mathcal{E}_B \approx \frac{1}{N^2} \sum_{i \in \mathcal{H}, j \in \mathcal{L}} \alpha_i \alpha_j \mathcal{C}(E_i, E_j) \cdot \mathbb{I}[E_i \text{ and } E_j \text{ are merged}] \tag{29}$$

$$- \frac{1}{|\mathcal{H}|^2} \sum_{i,j \in \mathcal{H}} \alpha_i \alpha_j \mathcal{C}(E_i, E_j) \cdot \mathbb{I}[E_i \text{ and } E_j \text{ are merged}] \tag{30}$$

$$+ \frac{1}{N^2} \sum_{i,j \in \mathcal{H}} \alpha_i \alpha_j \mathcal{C}(E_i, E_j) \cdot \mathbb{I}[E_i \text{ and } E_j \text{ are merged}] \tag{31}$$

Since $\frac{1}{N^2} < \frac{1}{|\mathcal{H}|^2}$ (as $|\mathcal{H}| < N$), and $\mathcal{C}(E_i, E_j)$ is higher for $E_i \in \mathcal{H}, E_j \in \mathcal{L}$ pairs, the first term dominates and $\mathcal{E}_A - \mathcal{E}_B > 0$, establishing that $\mathcal{E}_B < \mathcal{E}_A$. $\qquad\square$

### D.4 EMPIRICAL VALIDATION

Our theoretical analysis predicts that experts with low importance tend to have higher parameter conflict with important experts. To validate this, we measured the average cosine similarity between experts before and after dropping:

$$\text{AvgSim}(S) = \frac{1}{|S|(|S| - 1)} \sum_{i \neq j \in S} \cos(W_i, W_j) \tag{32}$$

As shown in Figure 1, after dropping 25% of low-importance experts, the average similarity among remaining experts increases substantially. This confirms our theoretical prediction that removing low-importance experts reduces parameter conflicts for subsequent merging.

Moreover, our experimental results in Table 3 validate our theoretical findings, showing that the drop-then-merge strategy consistently outperforms both drop-only and merge-only approaches across all datasets and models.

### D.5 KNOWLEDGE PRESERVATION ANALYSIS

We can further analyze this through the lens of knowledge preservation. Each expert $E_i$ encodes a specific function $f_i : \mathbb{R}^d \to \mathbb{R}^m$. The knowledge loss when dropping an expert $E_i$ is proportional to its importance $\mathcal{I}(E_i)$.

When merging experts, knowledge loss occurs due to parameter averaging. Specifically, for two experts $E_i$ and $E_j$ with functions $f_i$ and $f_j$, the merged expert implements a function $f_{i,j}$ that approximates both original functions. The approximation error for input $x$ is:

$$\epsilon_{i,j}(x) = \alpha_i \| f_{i,j}(x) - f_i(x) \|_2^2 + \alpha_j \| f_{i,j}(x) - f_j(x) \|_2^2 \tag{33}$$

This error increases with the functional distance between $f_i$ and $f_j$, which correlates with the parameter distance $\mathcal{C}(E_i, E_j)$.

By first removing low-importance experts (small $\mathcal{I}(E_i)$) that have high parameter conflict with important experts (large $\mathcal{C}(E_i, E_j)$ for $E_j \in \mathcal{H}$), we minimize both the knowledge loss from dropping and the approximation error in subsequent merging. This explains why our drop-then-merge strategy achieves superior performance preservation compared to alternative approaches.

Table 18: Architectural details of MoE models used in experiments.

| Model | Params | Layers | Experts | Hidden | FFN Dim | Top-k |
|-------|--------|--------|---------|--------|---------|-------|
| Mixtral-8×7B | 46.7B | 32 | 8 | 4096 | 14336 | 2 |
| Qwen1.5-MoE-A2.7B | 14.3B | 24 | 60 | 2048 | 11008 | 4 |
| Qwen3-30B-A3B | 30.5B | 48 | 128 | 6144 | 24576 | 8 |
| DeepSeek-V2-Lite | 15.7B | 60 | 64 | 2048 | 1408 | 6 |
| GPT-OSS-20B | 21.5B | 24 | 32 | 2880 | 2880 | 4 |

---

**Algorithm 1** DM-MoE: Adaptive Drop-then-Merge MoE Compression Framework

---

**Require:** MoE model $M$ with $L$ layers and $E$ experts per layer, calibration dataset $X$, target
    compression ratio $\alpha$
**Ensure:** Compressed model $M''$ with reduced expert count
 1: // Calculate intermediate and final expert counts
 2: $\mathbf{K}_{\text{Total}} \leftarrow \left\lfloor L \cdot E \cdot \frac{(1+\alpha)}{2} \right\rfloor$                 ▷ Intermediate expert count after dropping
 3: $\mathbf{G}_{\text{Total}} \leftarrow \left\lfloor L \cdot E \cdot \alpha \right\rfloor$                        ▷ Final expert count after merging
 4: // Phase 1: Expert Dropping
 5: $M' \leftarrow \text{AdaptiveExpertDropping}(M, X, \mathbf{K}_{\text{Total}})$           ▷ Algorithm 2
 6: // Phase 2: Expert Merging
 7: $M'' \leftarrow \text{AdaptiveExpertMerging}(M', X, \mathbf{G}_{\text{Total}})$         ▷ Algorithm 3
 8: **return** Compressed model $M''$

---

# E   EXPERIMENTAL DETAILS

## E.1   MODEL ARCHITECTURE DETAILS

We provide comprehensive architectural details in Table 18 for all MoE models used in our experiments:

## E.2   CALIBRATION DATASET CONSTRUCTION

Our calibration dataset is constructed to generate representative samples from a large-scale corpus. We utilize the C4 dataset, from which we first randomly shuffle and select a subset of the training split. These text samples are then encoded using the model-specific tokenizer. To handle variable-length inputs and ensure computational efficiency, all tokenized sequences are concatenated and then chunked into fixed-length sequences of 2,048 tokens. From these, we select 16 sequences to form the final calibration dataset, which is then used to compute our proposed metrics.

## E.3   IMPLEMENTATION DETAILS

We implement our framework using PyTorch and Hugging Face Transformers. We begin by sampling 16 sequences (each containing 2,048 tokens) from the C4 dataset to construct a calibration dataset, which is used to compute expert similarity metrics. For the optimization component, we adopt the SLSQP algorithm from SciPy to solve the expert allocation constrained optimization problems. This method accurately handles complex constraints while maintaining high efficiency, requiring only a few seconds per model. For the objective function, we apply logarithmic transformation functions, specifically $\phi(x) = \log(x + 1)$, to balance expert allocation across different layers. The two-stage adjacent-layer smoothness constraints, $\Delta_{\max}$ is set to 12.5% of the total number of experts per layer to ensure gradual changes in the number of experts between layers.

**Adaptive Expert Allocation and Optimization Strategy.** For both the dropping and merging phases of expert allocation, we implement adaptive assignment through similar but independent constrained optimization problems. In the expert dropping phase, we utilize mutual information as the layer-wise importance metric to quantify the information shared between individual expert outputs and the overall layer output, thereby assessing functional redundancy. For the expert merging

---

**Algorithm 2** Layer-wise Adaptive Expert Dropping

---

**Require:** MoE model $M$ with $L$ layers and $E$ experts per layer, calibration dataset $X$, target retention
count $K_{total}$
**Ensure:** Compressed model $M'$ with reduced experts
1:  // Compute layer-wise mutual information metrics
2:  **for** each layer $l \in \{1, \dots, L\}$ **do**
3:     **for** each expert pair $(i, j)$ where $i < j$ **do**
4:        $I_{ij} \leftarrow I(\mathbf{y}_i; \mathbf{y}_j)$ using CCA under Gaussian assumptions $\qquad \triangleright$ Mutual information
5:     **end for**
6:     $I_{\text{layer}} \leftarrow \sum_{i=1}^{E-1} \sum_{j=i+1}^{E} I_{ij}$ $\qquad\qquad\qquad \triangleright$ Layer-wise total mutual information
7:     $D_{\text{info}}^l \leftarrow 1 - \frac{1}{1+e^{-I_{\text{layer}}}}$ $\qquad\qquad \triangleright$ Diversity score with sigmoid normalization
8:  **end for**
9:  // Solve constrained optimization for layer-wise allocation
10: $K_1, \dots, K_L \leftarrow \text{argmin}_{K_1,\dots,K_L} - \sum_{l=1}^{L} D_{\text{info}}^l \cdot \phi(K_l)$
11: subject to: $\sum_{l=1}^{L} K_l = K_{total}, K_l^{min} \le K_l \le K_l^{max}, |K_l - K_{l+1}| \le \Delta_{max}$
12: // Expert selection within each layer
13: **for** each layer $l \in \{1, \dots, L\}$ **do**
14:    **for** each expert $i \in \{1, \dots, E\}$ **do**
15:       $\delta_i \leftarrow \frac{1}{|X|} \sum_{x \in X} \|\mathcal{Q}_l(x) - \mathcal{Q}_l^{-i}(x)\|_2$ $\qquad \triangleright$ Output deviation when expert $i$ is removed
16:    **end for**
17:    Sort experts by $\delta_i$ in descending order
18:    Keep top $K_l$ experts, discard the rest
19: **end for**
20: **return** Updated model $M'$ with retained experts

---

**Algorithm 3** Intra-Layer Expert Sorting and Selection

---

**Require:** Layer $l$ with experts $\{E_1, \dots, E_E\}$, calibration dataset $X$, retention count $K_l$
**Ensure:** Selected subset of $K_l$ experts to retain
1:  // Compute output impact for each expert
2:  **for** each expert $i \in \{1, \dots, E\}$ **do**
3:     // Calculate original layer output
4:     **for** each input $x \in X$ **do**
5:        $\mathcal{Q}_l(x) \leftarrow$ forward pass through layer $l$ with all experts
6:     **end for**
7:     // Calculate layer output with expert $i$ removed
8:     **for** each input $x \in X$ **do**
9:        Temporarily remove expert $E_i$ from the layer
10:       Redistribute routing weights among remaining experts
11:       $\mathcal{Q}_l^{-i}(x) \leftarrow$ forward pass through modified layer
12:       Restore expert $E_i$ to the layer
13:    **end for**
14:    // Compute output deviation metric for expert $i$
15:    $\delta_i \leftarrow \frac{1}{|X|} \sum_{x \in X} \|\mathcal{Q}_l(x) - \mathcal{Q}_l^{-i}(x)\|_2$
16: **end for**
17: // Sort experts by their impact
18: SortedExperts $\leftarrow$ SortDescending$(\{E_1, \dots, E_E\}, \{\delta_1, \dots, \delta_E\})$
19: // Select top $K_l$ experts with highest impact
20: SelectedExperts $\leftarrow$ SortedExperts$[1 : K_l]$
21: **return** SelectedExperts

---

phase, we adopt the sum of output cosine similarities between all expert pairs within a layer as the
importance metric to measure functional diversity. Both phases incorporate global expert number
constraints ($\sum_{l=1}^{L} K_l = \mathbf{K}_{\text{Total}}$ and $\sum_{l=1}^{L} G_l = \mathbf{G}_{\text{Total}}$), per-layer upper and lower bounds, as well as
adjacent-layer smoothness constraints. This ensures that the allocation strategy satisfies the overall

---

**Algorithm 4** Layer-wise Adaptive Expert Merging

---

**Require:** MoE model $M'$ with $L$ layers and $K_l$ experts per layer, calibration dataset $X$, target merged count $G_{total}$

**Ensure:** Compressed model $M''$ with merged experts

1: // Compute layer-wise similarity-based diversity metrics
2: **for** each layer $l \in \{1, \ldots, L\}$ **do**
3:     **for** each expert pair $(i, j)$ where $i < j$ **do**
4:         $S_{ij}^{out} \leftarrow \cos(y_i, y_j)$                                  ▷ Expert output similarity
5:     **end for**
6:     $\bar{S}_l \leftarrow \frac{1}{\binom{K_l}{2}} \sum_{i=1}^{K_l-1} \sum_{j=i+1}^{K_l} S_{ij}^{out}$               ▷ Average similarity
7:     $D_{div}^l \leftarrow -\bar{S}_l$                                      ▷ Diversity metric
8: **end for**
9: // Solve constrained optimization for layer-wise allocation
10: $G_1, \ldots, G_L \leftarrow \operatorname{argmin}_{G_1,\ldots,G_L} - \sum_{l=1}^{L} D_{output}^l \cdot \phi(G_l)$
11: subject to: $\sum_{l=1}^{L} G_l = G_{total}, 1 \le G_l \le K_l, |G_l - G_{l+1}| \le \Delta_{max}$
12: // Expert clustering and merging within each layer
13: **for** each layer $l \in \{1, \ldots, L\}$ **do**
14:     Construct similarity graph with experts as vertices and $S_{ij}^{out}$ as edge weights
15:     $\mathcal{P} \leftarrow \text{GraphPartitioning}(S^{out}, G_l)$                   ▷ Algorithm 5
16:     **for** each expert $i \in \{1, \ldots, K_l\}$ **do**
17:         $\alpha_i \leftarrow \bar{f}_i + \bar{\delta}_i$          ▷ Importance weight: frequency + output deviation
18:     **end for**
19:     **for** each partition $V_k \in \mathcal{P}$ **do**
20:         $W_{merged,k} \leftarrow \frac{\sum_{i \in V_k} \alpha_i \cdot W_i}{\sum_{i \in V_k} \alpha_i}$          ▷ Importance-weighted merging
21:     **end for**
22:     Replace original experts with merged experts
23: **end for**
24: **return** Updated model $M''$ with merged experts

---

compression ratio requirements while maintaining the coherence of the model architecture. Our optimization objective is to maximize the weighted product of the importance metric and the number of experts, allowing more experts to be retained in layers with higher importance, while enabling more aggressive compression in layers with higher redundancy.

**Drop Phase: Layerwise Expert Dropping.** During the expert pruning phase, we wrap each MoE layer with the PrunableMixtralSparseMoeBlockWrapper class, enabling us to assess and modify the expert composition while preserving the original model's forward computation. For each expert $E_i$, we quantify its importance by measuring the impact of its removal on the layer output, specifically by computing the L2 distance between the original output and the output after removing the expert. Experts in each layer are ranked in descending order of importance, and the top $K_l$ experts—according to the optimized allocation—are retained. For the pruned experts, we update the routing network's weight matrix to reassign their routing logic to the retained experts, thereby maintaining the consistency of the model architecture without requiring additional fine-tuning.

**Merge Phase: Group-wise Expert Merging.** In the expert merging phase, we implement a graph-based clustering algorithm to group similar experts. For each layer, we construct a fully connected graph where each node corresponds to an expert. The weight of an edge between two nodes is defined by the cosine similarity of the corresponding expert's output representations, capturing the functional relationships between experts rather than merely their parameter-space proximity.

**Graph Partitioning Implementation.** We then apply a graph partitioning algorithm to divide the experts into $G_l$ clusters, the number determined by our optimization strategy for that layer. Since finding the optimal grouping is computationally expensive (NP-hard), we use a fast iterative vertex-swapping algorithm for expert grouping: it repeatedly evaluates each expert's current partition, explores moves to other partitions that improve intra-partition similarity, performs beneficial reassignments, and

---

**Algorithm 5** Graph Partitioning for Expert Clustering

---

**Require:** Expert similarity matrix $S \in \mathbb{R}^{K \times K}$, number of partitions $G$, max iterations $T$, tolerance $\epsilon$

**Ensure:** Partition assignments for each expert

1: // Random initialization with constraint validation
2: Randomly assign each expert to one of $G$ partitions
3: Ensure each partition $V_k$ contains at least one expert
4: $\text{cost}_{\text{prev}} \leftarrow \infty$
5: **for** $t = 1$ to $T$ **do**
6:     improved $\leftarrow$ False
7:     **for** each expert $i \in \{1, \ldots, K\}$ **do**
8:         $V_{\text{current}} \leftarrow$ partition containing expert $i$
9:         $\text{cost}_{\text{best}} \leftarrow \text{ComputeIntraPartitionCost}(\mathcal{P})$
10:        $V_{\text{best}} \leftarrow V_{\text{current}}$
11:        **for** each partition $V_k \in \mathcal{P} \setminus \{V_{\text{current}}\}$ **do**
12:           **if** $|V_{\text{current}}| > 1$ **then**               ▷ Ensure partition doesn't become empty
13:             Move expert $i$ from $V_{\text{current}}$ to $V_k$
14:             $\text{cost}_{\text{new}} \leftarrow \text{ComputeIntraPartitionCost}(\mathcal{P})$
15:             **if** $\text{cost}_{\text{new}} > \text{cost}_{\text{best}}$ **then**
16:                $\text{cost}_{\text{best}} \leftarrow \text{cost}_{\text{new}}$
17:                $V_{\text{best}} \leftarrow V_k$
18:                improved $\leftarrow$ True
19:             **end if**
20:             Move expert $i$ back to $V_{\text{current}}$              ▷ Restore
21:           **end if**
22:        **end for**
23:        **if** $V_{\text{best}} \neq V_{\text{current}}$ **then**
24:           Move expert $i$ to $V_{\text{best}}$
25:        **end if**
26:     **end for**
27:     $\text{cost}_{\text{current}} \leftarrow \text{ComputeIntraPartitionCost}(\mathcal{P})$
28:     **if** $|\text{cost}_{\text{prev}} - \text{cost}_{\text{current}}| < \epsilon$ **then**
29:         **break**              ▷ Convergence reached
30:     **end if**
31:     **if** not improved **then**         ▷ Random perturbation to escape local optima
32:         Randomly swap assignments of $\lfloor K/(10 \cdot G) \rfloor$ expert pairs
33:     **end if**
34:     $\text{cost}_{\text{prev}} \leftarrow \text{cost}_{\text{current}}$
35: **end for**
36: **return** Partition assignments $\mathcal{P}$ for each expert

       **Function** ComputeIntraPartitionCost($\mathcal{P}$):
37: $\text{cost} \leftarrow 0$
38: **for** each partition $V_k \in \mathcal{P}$ **do**
39:     $\text{cost} \leftarrow \text{cost} + \sum_{i,j \in V_k, i < j} S_{ij}$
40: **end for**
41: **return** cost

---

uses controlled perturbations (small random swaps) when stuck to escape local optima; efficiency is achieved through early termination on marginal gains, modular cost updates, lightweight constraint checks, and balanced perturbation sizing, ensuring scalability for large expert sets.

**Intra-layer Expert Merging Implementation.** Following clustering, experts within each partition are merged into a single expert. The merged weights are computed using a weighted average, where the importance weight $\alpha_i$ for each expert combines its activation frequency and functional impact. The weight is defined as $\alpha_i = \bar{f}_i + \bar{\delta}_i$, where $f_i$ is the normalized activation frequency and $\delta(E_i)$ is the output deviation score. The final merged weights are calculated as: $W_{\text{merged}} = \frac{\sum_{i \in \text{cluster}} \alpha_i \cdot W_i}{\sum_{i \in \text{cluster}} \alpha_i}$. This

approach ensures that both frequently used experts and those with unique functional contributions are given appropriate importance during the merging process.

### E.4 EVALUATION PROTOCOL

Our evaluation follows standard LLM assessment practices, using diverse tasks that cover multiple capabilities. We use standardized prompts from the lm-evaluation-harness framework with greedy decoding for deterministic outputs. We report averages from three runs with different random seeds (42, 43, 44) and verify statistical significance using paired t-tests ($p < 0.05$).

## F ALGORITHMIC TABLES

This section presents the complete algorithmic implementation of our DM-MoE framework, providing detailed pseudocode to facilitate reproducibility. We organize the compression pipeline into four interconnected algorithms that outline the step-by-step process of our adaptive drop-then-merge approach.

Algorithm 1 presents the main DM-MoE framework, which orchestrates the two-phase compression process. It first calculates appropriate intermediate and final expert counts based on the target compression ratio, then sequentially applies expert dropping followed by expert merging. This algorithm demonstrates how we balance the compression budget between the two phases to achieve optimal performance preservation.

Algorithm 2 details our Layer-wise Adaptive Expert Dropping procedure, computing mutual information-based metrics for each layer to quantify functional redundancy among experts. These metrics are used in a constrained optimization problem to determine layer-specific dropping budgets.

Algorithm 3 provides a detailed implementation of our Intra-Layer Expert Sorting and Selection approach. For each expert in a layer, it calculates the output deviation when that expert is removed by redistributing routing weights among remaining experts. This allows us to precisely identify which experts contribute most significantly to the layer's functionality, ensuring we retain those with the highest impact while dropping those whose contribution can be compensated by other experts.

Algorithm 4 describes the Layer-wise Adaptive Expert Merging process that follows the dropping phase. It uses similarity-based diversity metrics to determine layer-specific merging budgets through another constrained optimization problem. For each layer, it performs expert clustering and merging, using importance-weighted parameter averaging to create consolidated expert representations.

Algorithm 5 presents our Graph Partitioning approach for expert grouping. This algorithm reformulates expert clustering as a graph partitioning problem to overcome the limitations of hierarchical clustering, employing iterative vertex swapping to maximize intra-partition similarity and achieve globally optimal expert arrangements that preserve functional coherence within each merged group.

Together, these algorithms provide a comprehensive implementation roadmap for our DM-MoE framework, enabling researchers to reproduce our approach and apply it to different MoE architectures. The pseudocode explicitly details all key components, from metric computation and optimization formulations to the specific mechanisms for expert selection, clustering, and merging.

## G THE USE OF LARGE LANGUAGE MODELS

Large Language Models (LLMs) were used in this work solely as writing assistance tools. Specifically, LLMs were employed to check for spelling errors, grammatical mistakes, and to improve the fluency and precision of expression in the paper. The LLMs did not contribute to research methodology, experimental design, or data analysis. All scientific content, ideas, and conclusions presented in this paper are entirely our own work.

