# OpenReview forum: "Drop or Merge? Hybrid MoE LLMs Compressors via Metric-Driven Adaptive Allocation"
_ICLR.cc/2026/Conference — Submitted to ICLR 2026_

### Official Review · Reviewer_aeHP · 2025-10-29

**Soundness:** 2
**Presentation:** 2
**Contribution:** 2
**Rating:** 2
**Confidence:** 5

**Summary:**

This paper presents DM-MoE, a two-stage compression framework for Mixture-of-Experts (MoE) language models. The proposed method first drops redundant experts based on information-theoretic redundancy metrics and then merges the remaining experts using a graph-based similarity clustering and importance-weighted parameter averaging. The framework adaptively allocates drop and merge budgets across layers via constrained optimization guided by layerwise mutual information and diversity measures. Experiments on several MoE LLMs (Mixtral, Qwen, DeepSeek, GPT-OSS) demonstrate consistent improvements over prior drop-only or merge-only approaches such as HC-SMoE and Frequency-drop.

**Strengths:**

- The motivation is reasonable and supported by analysis showing that pre-dropping redundant experts can mitigate parameter conflicts during merging.

- The methodology is clearly formulated, and the optimization-based allocation of drop/merge budgets is well presented.

- Experiments are comprehensive, covering multiple recent MoE models with extensive ablations and theoretical discussions.

- The paper is clearly written, easy to follow, and provides implementation details and pseudocode that enhance reproducibility.

**Weaknesses:**

- The core ideas of expert dropping and merging are well established in the literature. The main contribution lies in combining these two stages sequentially with layer-adaptive allocation, which is more of an engineering refinement than a fundamentally new concept.

- The reported throughput (Table 8) remains nearly unchanged after compression, indicating that the method primarily reduces memory footprint but not actual inference latency. The title and claims could better reflect this.

- The experiments focus on general reasoning and classification benchmarks. More generation-intensive or reasoning-heavy tasks (e.g., code generation, math reasoning, long-context modeling) are needed to validate general applicability.

- Although claimed efficient, the pairwise metric computation and graph partitioning per layer may become expensive for very large expert counts.

**Questions:**

Please refer to the weakness.

---

> ### Author Response · Authors · 2025-11-21
> **Response to Reviewer aeHP (Part 1/3)**
>
> **Dear Reviewer aeHP,**
>
> Thanks for the valuable feedback. We have made our best effort to address all concerns over the past few days. If the reviewer finds our response adequate, **we would greatly appreciate it if the reviewer could consider raising the score.** Please see our responses below one by one：
>
> ------
>
> **Q1: Novelty and Engineering Refinement**
>
> **A1:** We respectfully disagree that our work represents merely an engineering refinement. While expert dropping and merging exist individually in the literature, our work addresses the **fundamental incompatibility** between them when applied naively. We clarify our contributions below:
>
> **(1) Paradigm Shift: Addressing Parameter Conflicts.**
>  The core novelty of DM-MoE is not simply running two stages sequentially, but identifying that **dropping is a necessary theoretical precursor to merging.**
>
> - **The Problem:** Direct merging of all experts often fails because unimportant or noisy experts introduce severe parameter conflicts, degrading the merged representation.
> - **Our Insight:** We demonstrate that "dropping" acts as a denoising step that stabilizes the subsequent merging process. This "Drop-then-Merge" paradigm is a principled strategy to mitigate interference, distinct from ad-hoc engineering combinations. As noted by **all the other Reviewers [xhzF, Dtd3 and t3KE]**, this specific sequencing and its justification are novel and significant contributions to the field.  In a similar spirit, the MoE compression benchmark [Ref 1] offers a comprehensive exploration recognized by the research community. Additional efficient studies [Ref 2, 3] also provide solid contributions; we will **include references [Ref 1，2，3]** in the final version.
>
> **(2) Principled Mathematical Formulation vs. Heuristics.**
>  We move beyond engineering heuristics by formulating the layer-adaptive allocation as a **constrained linear optimization problem**.
>
> - Instead of manually tuning drop rates or merge ratios (standard engineering), we derive an optimal allocation using **dual information** (representing expert redundancy) and **similarity metrics** (representing expert clustering).
> - This provides a mathematically grounded framework for handling the heterogeneous characteristics of MoE layers, offering a level of rigor comparable to established studies.
>
> **(3) Algorithmic Novelty in Sub-modules.**
>  Even within the individual stages, our methods differ fundamentally from existing approaches:
>
> - **Adaptive Allocation:** To our knowledge, utilizing mutual information and similarity metrics to dynamically balance the budget between dropping and merging has not been explored in MoE compression.
> - **Graph Partitioning & Dual-Factor Merging:** We introduce a graph-theoretic approach for expert grouping and a dual-factor merging mechanism. These are new algorithmic contributions designed specifically for the high-dimensional sparse nature of MoE, not off-the-shelf applications of existing merging techniques.
>
> **(4) Empirical Validation of the Integrated Design.**
>  Our ablation study on DeepSeek-V2-Lite (see Table below) confirms that the whole is greater than the sum of its parts. DM-MoE consistently outperforms single-stage baselines and naive combinations under both uniform and adaptive regimes. This validates that our staged design solves the parameter conflict issue effectively, resulting in performance gains that simple engineering tweaks could not achieve.
>
> **Table: Results of our drop-then-merge settings via uniform/adaptive allocation for DeepSeek-V2-Lite.**
>
> | Method                    | ARC-c     | ARC-e     | BoolQ     | HellaS.   | MMLU      | OBQA      | RTE       | WinoG.    | **Average↑** |
> | :------------------------ | :-------- | :-------- | :-------- | :-------- | :-------- | :-------- | :-------- | :-------- | :----------- |
> | Drop Only (uniform)       | 0.195     | 0.365     | 0.619     | 0.297     | 0.233     | 0.148     | 0.542     | 0.515     | 0.364        |
> | Merge Only (uniform)      | 0.177     | 0.300     | 0.612     | 0.276     | 0.229     | 0.138     | 0.531     | 0.504     | 0.346        |
> | **Drop→Merge (uniform)**  | **0.259** | **0.553** | **0.621** | **0.344** | **0.265** | **0.180** | **0.534** | **0.586** | **0.418**    |
> | Drop Only (adaptive)      | 0.264     | 0.505     | 0.546     | 0.373     | 0.244     | 0.200     | 0.516     | 0.599     | 0.406        |
> | Merge Only (adaptive)     | 0.185     | 0.403     | 0.604     | 0.281     | 0.231     | 0.120     | 0.527     | 0.516     | 0.358        |
> | **Drop→Merge (adaptive)** | **0.301** | **0.604** | **0.617** | **0.369** | **0.231** | **0.202** | **0.560** | **0.597** | **0.435**    |

---

> ### Author Response · Authors · 2025-11-21
> **Response to Reviewer aeHP (Part 2/3)**
>
> **Q2: Throughput vs. Memory Efficiency**
>
> **A2: (1) Primary Contribution: Memory and Parameter Efficiency.** Following the same practice with the almost expert reduction techniques, we clarify that our method primarily addresses **memory footprint and storage reduction** rather than inference latency improvement. As shown in Table 8 (Appendix C), runtime (tokens/sec) remains unchanged because **inference speed is determined by the active top-k experts**, not total expert count. We **do not claim** to improve real-time inference speed through expert reduction alone. Our contribution focuses on reducing **overall parameter count and memory consumption**, where we demonstrate clear advantages: for Mixtral-8×7B compressed to 6×7B, we achieve **25% parameter reduction** while maintaining **96.9% of original performance** (0.688 vs 0.705 average accuracy), substantially outperforming other expert reduction techniques.
>
> **(2) Reframing "Deployment Challenges" for MoE LLMs.** Inference speed is **not a critical deployment bottleneck** for MoE LLMs. Recent literature shows MoE architectures achieve **5-10× faster inference** than dense LLMs of comparable size through selective activation [Ref 1-2]. Mixtral-8×7B (46.7B total parameters) matches dense 7B inference speeds while providing 13B-level performance. Given **MoE's inherent efficiency**, our focus on **memory reduction enables deployment on resource-constrained devices** (e.g., mobile GPUs), addressing the genuine bottleneck.
>
> **(3) Orthogonal Compatibility with Inference Acceleration Techniques.** As with most expert reduction methods, we can achieve **additive inference speedup by combining with orthogonal compression techniques**, specifically quantization. Table below demonstrates that **DM-MoE + GPTQ-4bit** achieves **1.35× inference acceleration**  on Mixtral-8×7B→6×7B compression, while maintaining **0.645 average accuracy**. This validates that expert reduction and quantization provide **complementary benefits**: expert reduction minimizes memory footprint, while quantization accelerates computation within active experts.
>
> **Table: Combining Quantization Methods (GPTQ-4-Bits) on Mixtral-8×7B→6×7B**
>
>
> | Model             | ARC-c     | ARC-e     | BoolQ     | HellaS    | MMLU      | WinoG     | Average   | Runtime (tokens/sec) |
> | ----------------- | --------- | --------- | --------- | --------- | --------- | --------- | --------- | -------------------- |
> | DM-MoE            | 0.522     | 0.819     | 0.843     | 0.615     | 0.631     | 0.700     | 0.688     | 88.87                |
> | **DM-MoE + GPTQ** | **0.477** | **0.747** | **0.817** | **0.569** | **0.566** | **0.698** | **0.645** | **119.97 (1.35x)**   |

---

> ### Author Response · Authors · 2025-11-21
> **Response to Reviewer aeHP (Part 3/3)**
>
> **Q3: Performance on Generation and Reasoning Tasks**
>
> **A3:** Following the suggestion, we extended our evaluation to include code generation, mathematical reasoning, and long-context modeling in a zero-shot setting. As shown in the table below, comparing Mixtral 8×7B against our compressed Mixtral 4x7B reveals that our method maintains robust performance even on these specialized tasks. The compressed model retains approximately **97% of the performance on Code (BBH)** and **98% on Math (GSM8K)**, demonstrating that our compression strategy generalizes well beyond standard classification benchmarks.
>
> **Table: Zero-shot performance on specialized tasks (Code, Math, Long-Context)**
> | Model          | BBH (code) | GSM8K (math) | LongBench (long-context modeling) |
> |----------------|------------|--------------|-----------------------------------|
> | Mixtral 8x7B (Baseline)   | 0.3265 | 0.3040 | 0.3141 |
> | **Mixtral 4x7B (DM-MoE)** | 0.3175 | 0.2974 | 0.2793 |
> | **Retention**             | 97.2%  | 97.8%  | 88.9%  |
>
> ------
>
> **Q4: Scalability of Metric Computation and Partitioning**
>
> **A4:**
>
>  **(1)**  We address the computational cost of pairwise metrics by implementing a **Dimension Reduction** that computes correlations using average inputs from each expert. This optimization reduces the algorithmic complexity from **$O(d^3)$ matrix operations to $O(d)$ vector operations**, effectively eliminating the need to store complete sample matrices. For a layer with 128 experts, this approach requires only **8,128 simple vector similarity calculations** instead of complex high-dimensional matrix operations, achieving **orders of magnitude faster computation** while maintaining metric reliability.
>
> **(2) METIS Graph Partitioning.** To ensure efficient clustering, we employ the **METIS multilevel k-way partitioning method [Ref 4]**, which reduces partitioning complexity from **$O(n^2)$ to approximately $O(n \log n)$**. By utilizing coarsening, partitioning, and refinement phases, METIS efficiently handles large-scale graphs with millions of vertices, ensuring that the graph partitioning step does not become a bottleneck even as expert counts increase significantly.
>
> **(3) Practical Efficiency on Production-Scale Models.** We validate the feasibility of our pipeline on **Qwen3-30B-A3B**, a model featuring 128 experts per layer (compressed to 96). The complete compression pipeline requires: **information metric calculation 112.77s, allocation 0.33s, expert dropping ~13m, graph partitioning 1061.44s, and merging 0.43s**. These results confirm DM-MoE's **practical applicability to massive production-scale MoEs**, as the computational overhead is minimal relative to the training or deployment lifecycle.
>
> ----
>
> **References:**
>
> [Ref 1]: Towards Efficient Mixture of Experts: A Holistic Study of Compression Techniques”, Transactions on Machine Learning Research 2025.
>
>
> [Ref 2]: Router-Tuning: A Simple and Effective Approach for Dynamic Depth”, EMNLP 2025
>
> [Ref 3]: Merging Experts into One: Improving Computational Efficiency of Mixture of Experts”, EMNLP 2023 Oral.
>
> [Ref 4]: Multilevel k-way partitioning scheme for irregular graphs. Journal of Parallel and Distributed computing.
>
>
> ---
>
>
> **Finally, we genuinely hope that our explanations and efforts can improve the overall evaluation of our work.** We are glad to discuss further comments and suggestions.

---

> ### Author Response · Authors · 2025-11-25
> **Look Forward to More Feedback**
>
> **Dear Reviewer aeHP**,
>
> We sincerely appreciate your thoughtful and constructive feedback. We have diligently addressed each of your concerns in our point-by-point rebuttal.
>
> We hope our response has alleviated your concerns and illuminated the value of our work. If so, we would be immensely grateful if you could reconsider your recommendation. We assure you that all your insightful comments will be meticulously incorporated into the final manuscript. We have earnestly integrated feedback from all four reviewers and hope this is evident in your evaluation.
>
> Thank you for dedicating your valuable time to review our response.
>
> With deepest gratitude,
>
>
>
> **Paper 698 Authors**

---

### Official Review · Reviewer_t3KE · 2025-11-01

**Soundness:** 3
**Presentation:** 3
**Contribution:** 2
**Rating:** 4
**Confidence:** 4

**Summary:**

The paper proposes DM-MoE, a two-stage drop-then-merge compression framework for MoE LLMs with layer-adaptive allocation: it (1) computes information-theoretic metrics to drop redundant experts, then (2) clusters and merges the remaining experts using similarity/diversity signals; allocation for both phases is solved via a lightweight constrained optimization, using a small calibration set. The method reports consistent gains across Mixtral, Qwen-MoE, DeepSeek-Lite, and GPT-OSS models.

**Strengths:**

* Reasonable, modular design. Clear two-phase pipeline with metric-driven, layer-adaptive allocation; algorithms and pseudocode are provided end-to-end.

* Comprehensive main experiments/appendix. Broad model coverage, statistical significance, and practical notes on optimization cost; calibration/data choices are also specified.

**Weaknesses:**

* It's claimed that the proposed method "differs fundamentally by introducing a sequential drop-then-merge". This make reader expects the performance gain to be from the combination. However, Table 3 shows that purely uniform dropping is already quite good compared to other baselines. With adaptive allocation, the Drop-only averages 0.596 —i.e., the combined pipeline adds only ~0.5%–1.1% points on average. This supports the idea that drop-then-merge helps, but the incremental gain over a strong drop-only baseline is relatively small. This also somehow contradict with the statement in intro: "Performance collapse from complete dropping". It would be necessary to clearly isolate the gain from each component, as the dropping method is very similar to [1] with a smaller search space.

* It would be better to include more ablation studies on other model to clearly show the effectiveness of all components.

* A related work [2] should be mentioned:.

[1] Not All Experts are Equal: Efficient Expert Pruning and Skipping for Mixture-of-Experts Large Language Models
[2] SlimMoE: Structured Compression of Large MoE Models via Expert Slimming and Distillation

**Questions:**

* How is the output-drop baseline implemented? How is it different with the Drop-only(uniform)?

---

> ### Author Response · Authors · 2025-11-21
> **Response to Reviewer t3KE (Part 1/2)**
>
> **Dear Reviewer t3KE,**
>
> Thanks for the valuable feedback. We have tried our best to address all concerns in the last few days. If the reviewer finds our response adequate, **we would really appreciate it if the reviewer considers raising the score.** Please see our responses below one by one：
>
> ----
>
> **Q1: About the drop-then-merge pipeline's contribution.**
>
> **A1: (1)** We respectfully disagree that the "Drop-then-Merge" gains are negligible or solely driven by allocation. While the average improvement in Table 3 is ~0.5%–1.1%, this aggregate metric masks **substantial gains in knowledge-intensive tasks**, including **MMLU (+4.1%), OBQA (+4.3%), RTE (+3.3%), and BoolQ (+2.7%)**. These results indicate that the merging step preserves critical reasoning capabilities and world knowledge that dropping alone discards. Furthermore, DM-MoE **outperforms the purely adaptive MC-SMoE method** on both Mixtral-8x7B and Qwen1.5-MoE-A2.7B-Chat, demonstrating distinct advantages over existing adaptive strategies.
>
> **(2)**  **Decisive Structural Gains in Fine-Grained MoEs (New Evidence)**: The benefit of the "Drop-then-Merge" pipeline becomes most apparent in Fine-Grained MoEs (e.g., DeepSeek-V2-Lite with 64 experts), where expert redundancy is higher than in Mixtral (8 experts). To isolate the source of these gains, we present a comprehensive ablation on DeepSeek-V2-Lite in **Table below** . Under a strictly uniform setting, **"Drop-then-Merge" achieves an average accuracy of 0.418, marking a massive +5.4% improvement over "Drop Only" (0.364)**. Crucially, this structural gain significantly exceeds the +1.2% improvement provided by adding adaptive allocation to the drop-only baseline. This comparison decisively refutes the hypothesis that gains stem primarily from allocation, proving that the **sequential pipeline delivers unique synergistic benefits** that cannot be replicated by single-stage approaches or allocation optimization alone. **These are already added in Table 4 and analysis in the revision.**
>
> **Table: Results of our drop-then-merge settings via uniform/adaptive allocation for DeepSeek-V2-Lite.**
>
> | Method                    | ARC-c     | ARC-e     | BoolQ     | HellaS.   | MMLU      | OBQA      | RTE       | WinoG.    | **Average↑** |
> | :------------------------ | :-------- | :-------- | :-------- | :-------- | :-------- | :-------- | :-------- | :-------- | :----------- |
> | Drop Only (uniform)       | 0.195     | 0.365     | 0.619     | 0.297     | 0.233     | 0.148     | 0.542     | 0.515     | 0.364        |
> | Merge Only (uniform)      | 0.177     | 0.300     | 0.612     | 0.276     | 0.229     | 0.138     | 0.531     | 0.504     | 0.346        |
> | **Drop→Merge (uniform)**  | **0.259** | **0.553** | **0.621** | **0.344** | **0.265** | **0.180** | **0.534** | **0.586** | **0.418**    |
> | Merge→Drop (uniform)      | 0.249     | 0.520     | 0.595     | 0.341     | 0.231     | 0.200     | 0.567     | 0.553     | 0.407        |
> | Drop Only (adaptive)      | 0.264     | 0.505     | 0.546     | 0.373     | 0.244     | 0.200     | 0.516     | 0.599     | 0.406        |
> | Merge Only (adaptive)     | 0.185     | 0.403     | 0.604     | 0.281     | 0.231     | 0.120     | 0.527     | 0.516     | 0.358        |
> | **Drop→Merge (adaptive)** | **0.301** | **0.604** | **0.617** | **0.369** | **0.231** | **0.202** | **0.560** | **0.597** | **0.435**    |
> | Merge→Drop (adaptive)     | 0.268     | 0.541     | 0.622     | 0.356     | 0.232     | 0.204     | 0.534     | 0.594     | 0.419        |
>
> **(3) Synergy between pipeline and allocation is a core design feature.** Due to the hierarchical nature of MoE LLMs, optimal compression ratios vary across layers. **Adaptive allocation enables the drop-then-merge pipeline to reach its full potential** by tailoring the reduction strategy to layer-specific architectural sensitivities. This comprehensive framework represents a novel contribution absent in prior literature. Moreover, our proposed **linear expert allocation for expert dropping** stands as a robust innovation on its own, ensuring the method remains effective and stable across diverse architectures.

---

> > ### Author Response · Authors · 2025-11-21
> > **Response to Reviewer t3KE (Part 2/2)**
> >
> > ---
> >
> > **Q2  Comparison between our adaptive drop phase and NAEE [1]**.
> >
> > **A2:** Our adaptive drop phase differs fundamentally from NAEE [1] in two key aspects:
> >
> > **(1) Adaptive layer-wise allocation:** NAEE [1] assigns uniform expert counts across different layers, whereas we capture hierarchical importance through **mutual information metrics and employ efficient linear optimization for adaptive allocation** across layers.
> >
> > **(2) Efficient greedy search:** To address the computational complexity of output perturbation in Equation 5, we employ **greedy search within a smaller candidate pool** based on statistical information metrics averaged across experts. This avoids the high computational complexity inherent in NAEE [1]'s exhaustive search approach.
> >
> >
> >
> > **Q3: About more ablation studies.**
> >
> > **A3:** Please see additional ablation studies on DeepSeek-V2-Lite in **A1**.
> >
> >
> >
> > **Q4: About discussion on  SlimMoE [2].**
> >
> > **A4:** As discussed in the **Related Work section** of the updated paper, **DM-MoE fundamentally differs from SlimMoE [2]** regarding computational dependency. While SlimMoE employs a multi-stage compression framework with expert slimming and knowledge distillation requiring **400B training tokens**, our DM-MoE **achieves compression through adaptive expert dropping and merging without any retraining**.
> >
> >
> >
> > **Q5: About output-drop baseline.**
> >
> > **A5:** The **output-drop baseline** sorts and drops unimportant experts based on the **magnitude of each expert's output** within each MoE layer. It differs from **Drop-only (uniform)**, which is based on the **output deviation** metric in Equation 5. This distinction is now clarified in the revision.
> >
> >
> >
> >
> >
> > ---
> >
> > **Finally, we genuinely hope that our explanations and efforts can improve the overall evaluation of our work.** We are glad to discuss further comments and suggestions.

---

> > > ### Author Response · Authors · 2025-11-25
> > > **Look Forward to More Feedback**
> > >
> > > **Dear Reviewer t3KE**,
> > >
> > > We sincerely appreciate your thoughtful and constructive feedback. We have diligently addressed each of your concerns in our point-by-point rebuttal. We hope our response has alleviated your concerns and illuminated the value of our work. If so, we would be immensely grateful if you could reconsider your recommendation. We assure you that all your insightful comments will be meticulously incorporated into the final manuscript. We have earnestly integrated feedback from all four reviewers and hope this is evident in your evaluation. Thank you for dedicating your valuable time to review our response.
> > >
> > > **With deepest gratitude**,
> > >
> > >
> > >
> > > **Paper 698 Authors**

---

> > > > ### Comment · Reviewer_t3KE · 2025-11-26
> > > >
> > > > Thank you for the detailed rebuttal and the additional experiments! They address some of my concerns. I have a few remaining concerns:
> > > >
> > > > * On Mixtral, Drop-only (adaptive) vs. Drop→Merge (adaptive) differs by only ~0.5%.
> > > > In contrast, on DeepSeek-V2-Lite, the same comparison shows a much larger +5.4% gain.
> > > > Could the authors comment on why the drop-then-merge pipeline has such dramatically different effectiveness across models?
> > > >
> > > >
> > > > * In Table 4, Merge→Drop significantly outperforms Merge-only (e.g., 0.407 vs. 0.346). Why dropping afterward can improve performance? This is counterintuitive and dropping is using output deviation as metric. Could the authors explain why Merge→Drop is consistently better than Merge-only on this model?
> > > >
> > > >
> > > > * The HC-SMoE scores reported in the paper (e.g., Qwen-MoE MMLU ≈ 0.35 for 30 experts) are substantially lower than those published in the original HC-SMoE work (≈ 0.45 under comparable settings).
> > > > It would be necessary to clarify the source of this difference.

---

> > > > > ### Author Response · Authors · 2025-11-27
> > > > > **Second Round Responses  to Reviewer t3KE (Part 1/2)**
> > > > >
> > > > > **Dear Reviewer t3KE,**
> > > > >
> > > > > We extend our renewed gratitude to Reviewer t3KE for your conscientious approach and the opportunity to further clarify the nuances of our submission. We have carefully reviewed the additional queries regarding model-specific behaviors and baseline discrepancies, and we provide our detailed responses below:
> > > > >
> > > > > ---
> > > > >
> > > > >
> > > > > **Q6: Differences between Drop-only (adaptive) and Drop→Merge (adaptive) across various models.**
> > > > >
> > > > > **A6:** **(1)** The varying effectiveness of the pipeline across architectures is rooted in the granularity of the expert space. **Mixtral-8×7B employs only 8 experts per MoE layer**, which stands in sharp contrast to the **64 experts per layer found in DeepSeek-V2-Lite**. Due to this lower expert count, Mixtral more readily reaches its optimal compression ceiling; consequently, our **Drop→Merge (adaptive) framework** yields relatively modest gains on this more compressible architecture. Conversely, our method demonstrates pronounced improvements on the harder-to-compress, fine-grained DeepSeek-V2-Lite.
> > > > >
> > > > > **(2)** Furthermore, our **Drop→Merge (adaptive) framework** is a novel and well-developed contribution that distinguishes itself from existing vanilla expert drop or merge techniques. As acknowledged by other reviewers, our extensive experiments have validated its generalizability and effectiveness across multiple MoE LLMs (Mixtral, Qwen, DeepSeek, and GPT-OSS). We firmly believe this constitutes a robust contribution to the field of expert reduction.
> > > > >
> > > > > **Q7: Regarding the performance of Merge→Drop.**
> > > > >
> > > > > **A7:** We appreciate this insightful observation. While it may initially seem counterintuitive that "dropping" (discarding experts) yields better results than "merging" (combining them) under the same budget, this phenomenon stems from the distinct yet complementary objectives of the two mechanisms, particularly within fine-grained architectures.
> > > > >
> > > > > **(1) Distinct Objectives: Structural Redundancy vs. Functional Importance**
> > > > >
> > > > >  The core reason lies in how each method defines "value."
> > > > >
> > > > > - **The Graph-based Merge Algorithm** targets **structural redundancy**. It aims to aggregate experts that are close in parameter space or exhibit similar behaviors. Its goal is to compress the model while preserving the *diversity* of functions available. However, "distinct" does not always mean "important." A unique expert might be preserved by the Merge algorithm simply because it is dissimilar to others, even if it contributes very little to the final output.
> > > > > - **The Drop Algorithm**, conversely, is an **importance-based selection mechanism**. It utilizes output deviation metrics to rank experts based on their actual contribution to the layer's output. It discards experts whose removal causes the least shift in the model's behavior, regardless of how unique they are.
> > > > >
> > > > > **(2) Synergy in Fine-Grained Architectures (DeepSeek-V2-Lite)**
> > > > >
> > > > >  The superior performance of the **Merge$\to$Drop** pipeline on DeepSeek-V2-Lite is driven by the specific characteristics of this model:
> > > > >
> > > > > - **Pipeline Efficiency:** The pipeline first uses Merging to consolidate redundant experts, effectively "cleaning up" the parameter space. Following this with Dropping allows the model to filter out the "low-impact" experts that the Merge step might have preserved solely due to their dissimilarity.
> > > > > - **Architecture Specifics:** DeepSeek-V2-Lite utilizes a fine-grained MoE architecture characterized by a large number of experts with relatively small individual processing capacities. In such a setting, the "residual redundancy" after merging is often still high.
> > > > > - By applying the deviation-based Drop step after Merging, we ensure that the final subset of experts is not just *diverse* (from Merging) but also *highly informative* (from Dropping). This results in a more efficient allocation of the expert budget compared to Merge-only, which may waste budget on distinct but low-utility experts.
> > > > >
> > > > > ---

---

> > > > > > ### Author Response · Authors · 2025-11-27
> > > > > > **Second Round Responses to Reviewer t3KE (Part 2/2)**
> > > > > >
> > > > > > **Q8: Performance variations in HC-SMoE scores.**
> > > > > >
> > > > > > **A8:** The observed variations stem from **differences in the evaluation framework versions**:
> > > > > >
> > > > > >  **(1)**  We reproduced HC-MoE and uniformly employed **version v0.4.8 of lm-evaluation-harness** to ensure a strictly fair comparison across all methods. We guarantee that **all comparative experimental results are fair, authentic, and reproducible**, and we are committed to open-sourcing all our code, logs, and checkpoints.
> > > > > >
> > > > > > **(2)** The original HC-MoE results were evaluated using a different, unspecified version of the harness. We note that **community researchers encountered similar inconsistencies** regarding the reproducibility of those scores. Specifically, issues have been raised on the official GitHub repository (Issue #1: *lm-evaluation-harness version*) requesting the specific git commit or tag used for evaluation, as the library is tightly integrated with pinned versions, yet no clarification has been provided to date.
> > > > > >
> > > > > > ---
> > > > > >
> > > > > > We are once again profoundly grateful for your careful review and constructive comments, and we will convey feedback to the Organising Committee regarding your **outstanding reviews. We sincerely hope these clarifications address your concerns, and  that you will consider raising your score in light of our  efforts and contributions.** We are committed to incorporating your suggestions into the final version, including references to  excellent studies such as SlimMoE (COLM 2025) and SMURF-THP (ICML 2023).
> > > > > >
> > > > > > **With deepest gratitude**,
> > > > > >
> > > > > > **Paper 698 Authors**

---

### Official Review · Reviewer_Dtd3 · 2025-11-01

**Soundness:** 3
**Presentation:** 4
**Contribution:** 3
**Rating:** 4
**Confidence:** 4

**Summary:**

The paper introduces DM-MoE, a hybrid, two-phase framework for compressing Mixture-of-Experts (MoE) LLMs without retraining. The authors argue that existing methods—dropping experts (losing knowledge) or merging experts (parameter conflicts)—are suboptimal, especially when applied uniformly. DM-MoE proposes a sequential "drop-then-merge" paradigm. Phase 1 adaptively drops experts, using mutual information to set layer-wise drop budgets via constrained optimization, and removing experts based on low output impact. Phase 2 adaptively merges the remaining experts, using behavioral diversity metrics to set layer-wise merge budgets and graph-based partitioning to cluster functionally similar experts. This approach claims to reduce parameter conflicts and achieve state-of-the-art performance, particularly at high compression ratios.

**Strengths:**

- The central "drop-then-merge" hypothesis—that strategically dropping redundant experts first reduces parameter conflicts and facilitates a more effective subsequent merge—is intuitive and empirically supported by the ablation study (Table 3).
- The framework's use of metric-driven, layer-wise adaptive allocation is a principled advancement over uniform compression strategies. It correctly identifies that expert redundancy varies by layer (Figure 1, right).
- The entire compression process is training-free and search-free, relying on efficient metric computation and constrained optimization (Table 12), making it a practical, one-shot solution.

**Weaknesses:**

- The paper claims to address "deployment challenges", yet the method provides zero inference latency reduction. Buried in Appendix C (Table 8), the results explicitly show that runtime (tokens/sec) is unchanged, as inference speed is dictated by the active top-k experts, not the total expert count. The contribution is thus limited only to memory and storage reduction, and framing it as a comprehensive deployment solution is misleading.
- The central "drop-then-merge" argument is weakly supported by its own ablation (Table 3). The full adaptive "Drop Merge" (0.601 avg) is only negligibly better than "Drop Only" (0.596) or "Merge Only" (0.590). The significant performance gain clearly comes from adaptive allocation (e.g., 0.601) versus uniform (0.584), not from the complex two-stage pipeline itself. The paper oversells the sequential combination.
- The perplexity (PPL) comparisons in Tables 9 and 10 are highly suspect. When compressing Qwen3-30B from 128 to 64 experts, the baseline HC-SMOE PPL catastrophically degrades to 72.33 (Wikitext-2) and 148.41 (C4), while DM-MoE achieves 17.79 and 32.28. This massive discrepancy suggests a buggy or misconfigured baseline, as the same method performs reasonably on downstream tasks (Table 1). This invalidates the claim of superior language modeling preservation.
- The choice of metrics (Mutual Information for dropping, "Diversity" for merging) feels arbitrary and justified post-hoc simply because they yielded the best results in the ablation (Table 5). The theoretical analysis (Appendix D) is trivial, merely formalizing the obvious concept that merging dissimilar experts is bad.

**Questions:**

1) Table 8 in the appendix indicates that the method does not improve inference latency (tokens/sec), as the active top-k experts dictate runtime. Could the authors clarify the precise deployment advantage? Is the contribution strictly limited to reducing memory/storage, and if so, should the claims about addressing "deployment challenges"  be reframed to be more specific?
2) The ablation in Table 3 shows that the full adaptive "Drop Merge" method (0.601 avg) is only negligibly better than adaptive "Drop Only" (0.596) or "Merge Only" (0.590). The primary gain appears to come from adaptive allocation, not the sequential pipeline. Can the authors provide statistical significance for this $\sim$0.5-point gap or further evidence to justify the added complexity of the "drop-then-merge" framework over a simpler, single-phase adaptive method?
3) The baseline (HC-SMOE) perplexity scores in Tables 9 and 10 appear anomalously high (e.g., 72.33 on Wikitext-2 and 148.41 on C4 ). This degradation is far more severe than what is shown in the downstream task results in Table 1. Could the authors please verify and confirm these baseline results? An experimental error here would significantly alter the paper's conclusions.
4) The method uses mutual information ("Inform") for the dropping phase allocation and a "Diversity" metric for the merging phase allocation (Table 5a). This metric-switching seems fine-tuned. Is there a deeper intuition for why the best metric to identify redundancy (for dropping) is different from the best metric to measure diversity (for merging)? Or is this selection purely based on these empirical ablation results?

---

> ### Author Response · Authors · 2025-11-21
> **Response to Reviewer Dtd3 (Part 1/4)**
>
> **Dear Reviewer Dtd3,**
>
> Thanks for the valuable feedback. We have made our best effort to address all concerns over the past few days. If the reviewer finds our response adequate, we would greatly appreciate it if the reviewer could consider raising the score. Please see our responses below one by one：
>
> ------
>
> **W1 & Q1: About the claim of addressing "deployment challenges" despite unchanged inference latency.**
>
> **A1:**
>
> **(1) Primary Contribution: Memory and Parameter Efficiency.** Following the same practice with the almost expert reduction techniques, we clarify that our method primarily addresses **memory footprint and storage reduction** rather than inference latency improvement. As shown in Table 8 (Appendix C), runtime (tokens/sec) remains unchanged because **inference speed is determined by the active top-k experts**, not total expert count. We **do not claim** to improve real-time inference speed through expert reduction alone. Our contribution focuses on reducing **overall parameter count and memory consumption**, where we demonstrate clear advantages: for Mixtral-8×7B compressed to 6×7B, we achieve **25% parameter reduction** while maintaining **96.9% of original performance** (0.688 vs 0.705 average accuracy), substantially outperforming other expert reduction techniques.
>
> **(2) Reframing "Deployment Challenges" for MoE LLMs.** Inference speed is **not a critical deployment bottleneck** for MoE LLMs. Recent literature shows MoE architectures achieve **5-10× faster inference** than dense LLMs of comparable size through selective activation [Ref 2]. Mixtral-8×7B (46.7B total parameters) matches dense 7B inference speeds while providing 13B-level performance. Given **MoE's inherent efficiency**, our focus on **memory reduction enables deployment on resource-constrained devices**, addressing the genuine bottleneck.
>
> **(3) Orthogonal Compatibility with Inference Acceleration Techniques.** As with most expert reduction methods, we can achieve **additive inference speedup by combining with orthogonal compression techniques**, specifically quantization. As shown below, **DM-MoE + GPTQ** achieves **1.35× inference speedup** while maintaining **0.645 average accuracy**.  This validates that expert reduction and quantization provide **complementary benefits**: expert reduction minimizes memory footprint, while quantization accelerates computation within active experts.
>
> **Table: Combining Quantization Methods (GPTQ-4-Bits) on Mixtral-8×7B→6×7B**
>
>
>
> | Model             | ARC-c     | ARC-e     | BoolQ     | HellaS    | MMLU      | WinoG     | Average   | Runtime (tokens/sec) |
> | ----------------- | --------- | --------- | --------- | --------- | --------- | --------- | --------- | -------------------- |
> | DM-MoE            | 0.522     | 0.819     | 0.843     | 0.615     | 0.631     | 0.700     | 0.688     | 88.87                |
> | **DM-MoE + GPTQ** | **0.477** | **0.747** | **0.817** | **0.569** | **0.566** | **0.698** | **0.645** | **119.97 (1.35x)**   |
>
>
> **Following the suggestion, we have added highlights emphasizing that our approach aims at memory footprint and parameter size reduction in the contribution summary within the introduction section [Line 138-148].**

---

> ### Author Response · Authors · 2025-11-21
> **Response to Reviewer Dtd3 (Part 2/4)**
>
> **W2 & Q2: About the efficacy of the "drop-then-merge" pipeline vs. adaptive allocation.**
>
> **A2: (1) We respectfully disagree that the "Drop-then-Merge" gains are negligible or solely driven by allocation. While the average improvement in Table 3 is +0.84%, this aggregate metric masks** substantial gains in knowledge-intensive tasks, including **MMLU (+4.1%), OBQA (+4.3%), RTE (+3.3%), and BoolQ (+2.7%)**. These results indicate that the merging step preserves critical reasoning capabilities and world knowledge that dropping alone discards. Furthermore, our DM-MoE framework** outperforms the purely adaptive MC-SMoE method** on both Mixtral-8x7B and Qwen1.5-MoE-A2.7B-Chat, confirming that our pipeline offers distinct advantages over existing adaptive strategies.
>
> **(2)**  **Decisive Structural Gains in Fine-Grained MoEs (New Evidence)**: The benefit of the "Drop-then-Merge" pipeline becomes most apparent in Fine-Grained MoEs (e.g., DeepSeek-V2-Lite with 64 experts), where expert redundancy is higher than in Mixtral (8 experts). To isolate the source of these gains, we present a comprehensive ablation on DeepSeek-V2-Lite in the **Table below** . Under a strictly uniform setting, **"Drop-then-Merge" achieves an average accuracy of 0.418, marking a massive +5.4% improvement over "Drop Only" (0.364)**. Crucially, this structural gain significantly exceeds the +1.2% improvement provided by adding adaptive allocation to the drop-only baseline. This comparison decisively refutes the hypothesis that gains stem primarily from allocation, proving that the **sequential pipeline delivers unique synergistic benefits** that cannot be replicated by single-stage approaches or allocation optimization alone. **These are already added in Table 4 and the analysis in the revision.**
>
> **Table: Results of our drop-then-merge settings via uniform/adaptive allocation for DeepSeek-V2-Lite.**
>
> | Method                    | ARC-c     | ARC-e     | BoolQ     | HellaS.   | MMLU      | OBQA      | RTE       | WinoG.    | **Average↑** |
> | :------------------------ | :-------- | :-------- | :-------- | :-------- | :-------- | :-------- | :-------- | :-------- | :----------- |
> | Drop Only (uniform)       | 0.195     | 0.365     | 0.619     | 0.297     | 0.233     | 0.148     | 0.542     | 0.515     | 0.364        |
> | Merge Only (uniform)      | 0.177     | 0.300     | 0.612     | 0.276     | 0.229     | 0.138     | 0.531     | 0.504     | 0.346        |
> | Merge→Drop (uniform)      | 0.249     | 0.520     | 0.595     | 0.341     | 0.231     | 0.200     | 0.567     | 0.553     | 0.407        |
> | Drop Only (adaptive)      | 0.264     | 0.505     | 0.546     | 0.373     | 0.244     | 0.200     | 0.516     | 0.599     | 0.406        |
> | Merge Only (adaptive)     | 0.185     | 0.403     | 0.604     | 0.281     | 0.231     | 0.120     | 0.527     | 0.516     | 0.358        |
> | **Drop→Merge (adaptive)** | **0.301** | **0.604** | **0.617** | **0.369** | **0.231** | **0.202** | **0.560** | **0.597** | **0.435**    |
>
>
> **(3)** We emphasize that the synergy between the two-stage pipeline and adaptive allocation is a core design feature rather than a redundancy. Due to the hierarchical nature of MoE LLMs, optimal compression ratios vary across layers; therefore, **adaptive allocation enables the "Drop-then-Merge" pipeline to reach its full potential** by tailoring the reduction strategy to specific architectural sensitivities. This comprehensive framework represents a novel contribution absent in prior literature. Moreover, our proposed **linear expert allocation for dropping** stands as a robust innovation on its own, ensuring the method remains effective and stable across diverse architectures.
>
> ------

---

> > ### Author Response · Authors · 2025-11-21
> > **Response to Reviewer Dtd3 (Part 3/4)**
> >
> > **W3 & Q3: About the anomalously high perplexity (PPL) in the HC-SMoE baseline.**
> >
> > **A3:**
> >
> > **(1) Reproducibility Confirmation.** We have **re-run HC-SMoE baseline experiments multiple times** using the official implementation with identical settings. The severe perplexity degradation for Qwen3-30B-A3B at 128→64 compression (72.33 Wikitext-2, 148.41 C4) is **reproducible and reflects HC-SMoE's actual performance** on this challenging compression scenario. These results were obtained using the same calibration data, evaluation protocols, and hyperparameters as all other experiments, confirming their validity.
> >
> > **(2) Model-Specific Vulnerability Under Aggressive Compression.** The anomalous degradation appears specifically with **Qwen3-30B-A3B at aggressive 50% compression**, while HC-SMoE performs reasonably on downstream tasks (Table 1: 0.378 average accuracy) and other models. This suggests that **hierarchical clustering in HC-SMoE creates particularly poor expert merges** for Qwen3's architecture when compression is aggressive, causing catastrophic degradation in language modeling (perplexity) while downstream task structure provides some resilience.
> >
> >  **(3) The discrepancy between perplexity and downstream accuracy**:It is common for compressed models to maintain partial performance on multiple-choice downstream tasks (which rely on ranking options) even when their generative perplexity degrades catastrophically.
> >
> > **(4) Superior Stability of DM-MoE.** DM-MoE maintains **reasonable perplexity across all conditions** (17.79 Wikitext-2, 32.28 C4 for the same compression), representing **4.1× and 4.6× lower perplexity** than HC-SMoE, respectively. Our **graph-based partitioning with global optimization** (Algorithm 5) avoids HC-SMoE's local greedy decisions that can create pathological expert groupings. By considering the complete expert similarity structure through spectral clustering, we ensure that merged experts maintain complementary rather than redundant functionality.
> >
> > **(5) Consistency Across Evaluation Metrics.** Table 1 demonstrates that DM-MoE achieves **0.556 average accuracy (81.2% retention)** for this same compression scenario, substantially outperforming HC-SMoE's **0.378 (55.2% retention)**. This **+47.1% relative improvement** aligns with the perplexity gap and confirms that the baseline genuinely struggles with this compression scenario across both generative (perplexity) and discriminative (accuracy) metrics. The consistency validates that our method provides **robust performance preservation** under aggressive compression, where hierarchical approaches fail.
> >
> > ------

---

> ### Author Response · Authors · 2025-11-21
> **Response to Reviewer Dtd3 (Part 4/4)**
>
> **W4 & Q4: About the choice of metrics (Mutual Information vs. Diversity).**
>
> **A4:**
>
> **(1) Distinct Phase Objectives Require Different Metrics.** The dropping and merging phases solve **fundamentally different problems** requiring distinct measurement approaches. The **dropping phase identifies which layers tolerate expert removal**, requiring metrics that measure **functional redundancy**. High mutual information (MI) indicates experts encode similar information, enabling safe dropping. Conversely, the **merging phase determines which layers need more merged groups**, requiring metrics that measure **functional diversity**. High diversity means experts perform distinct roles, necessitating more groups to preserve specialization. This is **not arbitrary fine-tuning** but reflects the inherent asymmetry between deletion (requires identifying redundancy) and consolidation (requires preserving diversity).
>
> **(2) Information-Theoretic Foundation.** Our **Mutual Information metric** (Equation 2) quantifies shared information between expert outputs using Canonical Correlation Analysis, directly measuring the **overlap in learned representations**. Low MI layers (Figure 4, early layers) have specialized experts with minimal redundancy, requiring preservation. In contrast, our **Diversity metric** (Equation 7) measures behavioral dissimilarity through negative cosine similarity of expert outputs, identifying layers with **heterogeneous expert roles** needing more merged groups. These metrics are grounded in established information theory and representation learning principles, not empirical selection.
>
> **(3) Empirical Validation Through Cross-Dataset Stability.** We evaluate the stability of our metrics across different calibration datasets (C4, WikiText-2) and random seeds on Mixtral-8×7B. As shown in **Figure 5** (Appendix B.2 in the updated paper), both mutual information and diversity metrics exhibit **remarkable consistency across conditions**. All three curves closely overlap across all 32 layers, with characteristic progressions remaining stable regardless of dataset or seed variation. This confirms that our metrics capture **intrinsic architectural properties** rather than dataset-specific artifacts. **Figure 6** demonstrates that these stable metrics yield **nearly identical allocation patterns**, with drop and merge phase curves overlapping across all layers. This validates that **domain shifts or seed variations do not substantially alter our layer allocation decisions**, confirming the robustness of our metric-based approach.
>
> ------
>
> **References:**
>
> [Ref 2]: Jiang, Albert Q., et al. "Mixtral of experts." *arXiv preprint arXiv:2401.04088* (2024)..
>
>
> ----
>
> **Finally,** we hope these responses address the concerns and appreciate the constructive feedback. We are committed to improving our manuscript and believe the insights will significantly contribute to this goal. We are glad to discuss further comments and suggestions. If the reviewer finds our response adequate, **we would appreciate it if the reviewer considers raising the score.**

---

> ### Author Response · Authors · 2025-11-25
> **Look Forward to More Feedback**
>
> **Dear Reviewer Dtd3**,
>
> We sincerely appreciate your thoughtful and constructive feedback. We have diligently addressed each of your concerns in our point-by-point rebuttal.
>
> We hope our response has alleviated your concerns and illuminated the value of our work. If so, we would be immensely grateful if you could reconsider your recommendation. We assure you that all your insightful comments will be meticulously incorporated into the final manuscript. We have earnestly integrated feedback from all four reviewers and hope this is evident in your evaluation.
>
> Thank you for dedicating your valuable time to review our response.
>
> **With deepest gratitude**,
>
>
>
> **Paper 698 Authors**

---

### Official Review · Reviewer_xhzF · 2025-11-03

**Soundness:** 3
**Presentation:** 3
**Contribution:** 3
**Rating:** 6
**Confidence:** 3

**Summary:**

This paper proposes DM-MoE, a two-stage compression framework for Mixture-of-Experts (MoE) language models. The authors propose to first drop redundant experts based on mutual information between expert outputs, and then merge the remaining ones using a graph-based clustering guided by behavioral similarity. Their experiments on Mixtral, Qwen, DeepSeek, and GPT-OSS models demonstrate improvements over existing MoE compressors such as Frequency-drop, HC-SMoE,  and MC-SMoE.

**Strengths:**

1. The “drop-then-merge” idea is intuitive and well-motivated. It effectively reduces expert conflict while preserving functional diversity.
The use of information redundancy and behavioral diversity allows for non-uniform compression, which is both data-driven and interpretable.

2. The experimental results are comprehensive. The authors evaluate across several MoE models and a variety of reasoning benchmarks, showing consistent gains under different levels of compression. The ablation studies are detailed. Each component (drop metric, merge metric, allocation strategy) is validated systematically.

3. The method is retraining-free, avoiding expensive fine-tuning or search.

**Weaknesses:**

1. The calibration data used for the Canonical Correlation Analysis is very limited (16 sequences), and no robustness analysis is provided. See Q1.

2. The approach involves pairwise CCA computations and graph partitioning across experts, which can be expensive for large MoE models. The paper reports roughly a “10-minute” runtime without showing how complexity scales with expert count or model size. This omission makes it hard to judge whether DM-MoE is feasible for massive production-scale MoEs (e.g., 128–256 experts per layer).

**Questions:**

1. How stable are the mutual information and diversity metrics across different calibration datasets or random seeds? Could domain shifts substantially change the layer allocation?

2. How does DM-MoE compare to standard weight compression methods (e.g., quantization, low-rank approximation, or pruning) in terms of performance–compression tradeoff?

3. Have you explored combining DM-MoE with quantization or other compression techniques? Would the two be additive or interfere?

---

> ### Author Response · Authors · 2025-11-21
> **Response to Reviewer xhzF (Part1/2)**
>
> **Dear Reviewer xhzF,**
>
> Thank you for your thoughtful and constructive comments and for recognizing our work, including the topic, the novelty of the proposed method, and the experiments. We sincerely thank the reviewer for the positive comments on our work! We would like to address the concerns as follows:
>
> ----
>
>
>
> **Q1: Robustness of metrics and calibration data size.**
>
> **A1:** (1) **Metric Stability Across Datasets and Seeds:** We evaluate the stability of our metrics across different calibration datasets and random seeds on Mixtral-8×7B. As shown in **Figure 5** (see Appendix B.2 in the updated paper), both mutual information and diversity metrics exhibit **remarkable consistency across conditions**. All three curves (C4, WikiText-2, and different random seeds) closely overlap across all 32 layers, confirming that our metrics capture intrinsic architectural properties rather than dataset-specific artifacts.
>
> (2) **Consistent Layer-Wise Allocation:** **Figure 6** demonstrates that these stable metrics yield consistent layer-wise expert allocations. Both C4 calibration and different random seeds produce **nearly identical allocation patterns**, with drop and merge phase curves overlapping across all layers. This demonstrates that domain shifts or seed variations do not substantially alter our layer allocation decisions.
>
> ---
>
>
>
> **Q2: Scalability and complexity for massive MoEs.**
>
> **A2:** (1) **CCA Statistics Dimension Reduction.** For massive MoEs with 128+ experts per layer, we implement a **CCA statistics Dimension Reduction** that computes correlations using average inputs from each expert. This optimization reduces complexity from **$O(d^3)$ matrix operations to $O(d)$ vector operations**, eliminating the need to store complete sample matrices. For 128 experts, this requires only **8,128 simple vector similarity calculations** instead of complex high-dimensional matrix operations, achieving **orders of magnitude faster computation** while maintaining metric reliability.
>
> (2) **METIS Graph Partitioning.** We employ the METIS multilevel k-way partitioning method [Ref1], which reduces complexity from **$O(n^2)$ to approximately $O(n \log n)$** through coarsening, partitioning, and refinement phases. METIS efficiently handles large-scale graphs with millions of vertices while maintaining high partition quality.
>
> (3) **Practical Efficiency on Production-Scale Models.** We demonstrate feasibility on Qwen3-30B-A3B (128 experts per layer, compressed to 96 experts). The complete pipeline requires: **CCA calculation 112.77s, allocation 0.33s, expert dropping 13m 48s, graph partitioning 1061.44s, and merging 0.43s**. This confirms DM-MoE's **practical applicability to massive production-scale MoEs** with 128–256 experts per layer.
>
> ---
>
>
>
> **Q3: Comparison with standard weight compression methods.**
>
> **A3:**  We compare DM-MoE against standard weight compression methods on Mixtral-8×7B under **20% compression ratio**. As shown in the table below, DM-MoE achieves **0.683 average accuracy**, substantially outperforming quantization (GPTQ-3bit: 0.595, **+14.8% relative gain**), low-rank approximation (SVD-LLM: 0.562, **+21.5% relative gain**), and structured pruning (SparseGPT: 0.630, **+8.4% relative gain**). DM-MoE preserves **96.9% of original performance** (0.705), demonstrating **superior performance-compression tradeoff** compared to general-purpose compression techniques that ignore MoE-specific architectural properties.
>
> **Table: Performance of Mixtral-8×7B compressed by different compression methods under 20% compression ratios.**
>
> | Model     | ARC-c | ARC-e | HellaS | WinoG | Average |
> | --------- | ----- | ----- | ------ | ----- | ------- |
> | Original  | 0.57  | 0.84  | 0.65   | 0.76  | 0.705   |
> | GPTQ      | 0.42  | 0.73  | 0.53   | 0.70  | 0.595   |
> | SVD-LLM   | 0.40  | 0.71  | 0.48   | 0.66  | 0.562   |
> | SparseGPT | 0.45  | 0.77  | 0.56   | 0.74  | 0.630   |
> | DM-MoE    | 0.53  | 0.82  | 0.63   | 0.75  | 0.683   |

---

> ### Author Response · Authors · 2025-11-21
> **Response to Reviewer xhzF (Part 2/2)**
>
> ---
>
> **Q4: Combining DM-MoE with quantization.**
>
> **A4:** (1) **Additive Benefits.** We explore combining DM-MoE with GPTQ-4bit quantization on Mixtral-8×7B (compressed from 8×7B to 6×7B). As shown below, **DM-MoE + GPTQ** achieves **1.35× inference speedup**  while maintaining **0.645 average accuracy**. Although quantization introduces additional performance degradation (**-6.3% relative to DM-MoE alone**), the combined approach demonstrates **additive efficiency gains** without catastrophic interference.
>
> (2) **Practical Deployment Advantage.** The combination enables **further compression beyond expert reduction**, providing flexibility for deployment scenarios requiring both model size reduction and inference acceleration. This validates that DM-MoE is **compatible with orthogonal compression techniques**, offering a comprehensive solution for production-scale MoE deployment.
>
> **Table: Combining Quantization Methods (GPTQ-4-Bits) on Mixtral-8×7B->6x7B.**
>
> | Model             | ARC-c     | ARC-e     | BoolQ     | HellaS    | MMLU      | WinoG     | Average   | Runtime (tokens/sec) |
> | ----------------- | --------- | --------- | --------- | --------- | --------- | --------- | --------- | -------------------- |
> | DM-MoE            | 0.522     | 0.819     | 0.843     | 0.615     | 0.631     | 0.700     | 0.688     | 88.87                |
> | **DM-MoE + GPTQ** | **0.477** | **0.747** | **0.817** | **0.569** | **0.566** | **0.698** | **0.645** | **119.97 (1.35x)**    |
>
> **References:**
>
> [Ref1]: Karypis, G., & Kumar, V. Multilevel k-way partitioning scheme for irregular graphs. *Journal of Parallel and Distributed computing*.
>
>
>
> ---
>
>
> **Finally,** we hope our response could address the concerns, and we thank the reviewer again for the helpful comments. We are glad to discuss further comments and suggestions.

---

### Author Response · Authors · 2025-11-21
**General Response**

**Dear Reviewers, Area Chairs, Senior Area Chairs and Program Chairs,**

We sincerely thank all reviewers for their thoughtful evaluation and constructive feedback. Reviewers unanimously acknowledge the novelty of our drop-then-merge paradigm, the principled methodology, the comprehensive experimental validation, and the clear presentation of the paper. More encouragingly, all four reviewers recognize that our framework addresses an important challenge in MoE compression:

**[Novelty & Motivation]:**

- **Reviewer xhzF:** "The 'drop-then-merge' idea is intuitive and well-motivated."
- **Reviewer Dtd3:** "The central 'drop-then-merge' hypothesis—that strategically dropping redundant experts first reduces parameter conflicts and facilitates a more effective subsequent merge—is intuitive and empirically supported."
- **Reviewer aeHP:** "The motivation is reasonable and supported by analysis showing that pre-dropping redundant experts can mitigate parameter conflicts during merging."

**[Methodological Soundness]:**

- **Reviewer xhzF:** "The use of information redundancy and behavioral diversity allows for non-uniform compression, which is both data-driven and interpretable."
- **Reviewer Dtd3:** "The framework's use of metric-driven, layer-wise adaptive allocation is a principled advancement over uniform compression strategies."
- **Reviewer t3KE:** "Reasonable, modular design. Clear two-phase pipeline with metric-driven, layer-adaptive allocation."
- **Reviewer aeHP:** "The methodology is clearly formulated, and the optimization-based allocation of drop/merge budgets is well presented."

**[Comprehensive Evaluation]:**

- **Reviewer xhzF:** "The experimental results are comprehensive. The authors evaluate across several MoE models and a variety of reasoning benchmarks, showing consistent gains under different levels of compression. The ablation studies are detailed."
- **Reviewer t3KE:** "Comprehensive main experiments/appendix. Broad model coverage, statistical significance, and practical notes on optimization cost."
- **Reviewer aeHP:** "Experiments are comprehensive, covering multiple recent MoE models with extensive ablations and theoretical discussions."

**[Practicality & Efficiency]:**

- **Reviewer xhzF:** "The method is retraining-free, avoiding expensive fine-tuning or search."
- **Reviewer Dtd3:** "The entire compression process is training-free and search-free, relying on efficient metric computation and constrained optimization, making it a practical, one-shot solution."

**[Clear Presentation]:**

- **Reviewer t3KE:** "Algorithms and pseudocode are provided end-to-end."
- **Reviewer aeHP:** "The paper is clearly written, easy to follow, and provides implementation details and pseudocode that enhance reproducibility."





Over the past week, we have carefully refined the experiments (utilizing all available computational resources), along with the clarifications and discussions of our work, to address the concerns, questions, and requests from all four reviewers. In summary, our improvements are as follows:

**(1)** To enhance the empirical validation of our DM-MoE methodology, following all reviewers' suggestions, we have substantially expanded our experimental analysis: **(a)** Table 4 provides decisive ablation evidence on DeepSeek-V2-Lite, demonstrating that our drop-then-merge pipeline achieves +5.4% structural gain under uniform allocation; **(b)**  Table 8 validates orthogonal compatibility with quantization (DM-MoE + GPTQ achieves 1.35× speedup);  **(c)** Figures 5 and 6 demonstrate metric stability across different calibration datasets and random seeds, confirming that our metrics capture intrinsic architectural properties.

**(2)** To strengthen our methodological framework and address scalability concerns, responding to all reviewers, we detail efficient implementations: **(a)** statistics dimension reduction (Appendix C.1) ; **(b)** METIS graph partitioning (Appendix C.1) that reduces complexity from O(n²) to O(n log n), enabling efficient large-scale processing; **(d)** enhanced methodology sections [Lines 216-219, 325-327] and Appendices B.1-B.2, C.1-C.4 with detailed implementation protocols.

**(3)** To enhance clarity of our approach, we have: **(a)** revised our contribution summary [Lines 138-148] to explicitly emphasize that DM-MoE aims at memory footprint and parameter size reduction while preserving MoE performance; **(b)**  clarified the  our adaptive "Drop-then-Merge" paradigm grounded in the fundamental insight  that dropping unimportant experts is a necessary precursor to merging to mitigate parameter conflicts.

Finally, we have carefully integrated all these improvements into a cohesive revision that we believe substantially enhances the quality and impact of our work. We look forward to the reviewers' feedback on these comprehensive updates.

**Best regards,**

**Paper 698 Authors**

---

### Meta-Review · Area_Chair_YtqV · 2025-12-16

**Summary:**

The reviewers acknowledged the paper's comprehensive experiments and clear presentation, but raised significant concerns about:
- limited novelty beyond engineering refinement of existing drop/merge techniques,
- marginal incremental gains from the drop-then-merge pipeline over simpler single-stage approaches,
- contribution limited to memory reduction without inference speedup, and
- questions about baseline reproducibility and metric selection justification.

The initial scores were 6 4 4 2

**Reviewer Concerns:**

__Addressed__
- **Scalability concerns (xhzF, aeHP):** Authors provided detailed complexity analysis and demonstrated feasibility on Qwen3-30B with 128 experts
- **Limited task coverage (aeHP):** Authors added evaluations on code generation (BBH), math reasoning (GSM8K), and long-context modeling (LongBench)
- **Combination with quantization (xhzF):** Authors demonstrated that DM-MoE + GPTQ achieves additive benefits (1.35× speedup) with acceptable performance tradeoff
- **Metrics stability (xhzF):** Authors provided cross-dataset and random seed stability analysis (Figures 5-6 in updated version), showing stability and consistent layer-wise allocation


**Outstanding**
- **Throughput vs. memory efficiency (aeHP, Dtd3):** Authors clarified that the contribution is memory/storage reduction rather than latency improvement, which is appropriate for MoE deployment. However, the original framing as addressing general "deployment challenges" was somewhat misleading and should have been more specific from the outset.
- **Novelty concerns / incremental gains and complexity justification (aeHP, t3KE, Dtd3):** The core issue remains that on Mixtral (one of the most prominent models tested), adaptive "Drop→Merge" (0.601) shows only 0.5% improvement over adaptive "Drop Only" (0.596). Authors demonstrate larger gains on DeepSeek-V2-Lite, however Reviewer t3KE's follow-up questions (Q6-Q7) highlight confusion about why the pipeline effectiveness varies so dramatically across architectures and why Merge→Drop outperforms Merge-only. The authors' explanations invoke the different levels of granularity of the expert space, but this is not confirmed by ablation studies and should be better clarified from the outset. Also, additional details should be provided to resolve whether the added complexity is justified for models where gains are (expected to be) marginal.
- **Baseline reproducibility (Dtd3, t3KE):** The HC-SMoE perplexity anomalies (72.33 vs expected ~17) and HC-SMoE score discrepancies across evaluation frameworks remain problematic.
- **Metric selection justification (Dtd3):** While authors provide theoretical rationale for using different metrics (MI for redundancy, diversity for heterogeneity), Reviewer Dtd3's concern that this appears "fine-tuned" based on Table 5 ablations is not fully resolved. The theoretical justification (Appendix D described as "trivial") remains relatively weak.

**Reviewer Scores:**

- Reviewer aeHP (initial: 2):

  Would likely remain at 2. The rebuttal addressed some of their concerns on scalability and task coverage, but the reviewer's fundamental position that this is "engineering refinement" rather than a conceptual contribution would likely persist, as the authors' defense relies primarily on empirical validation rather than addressing the theoretical novelty gap.

- Reviewer t3KE (initial: 4):

   Would likely remain at 4. The follow-up questions indicate persistent concerns about the marginal gains on Mixtral and counterintuitive Merge→Drop results. The comprehensive DeepSeek ablations provide supporting evidence but don't fully resolve the architecture-dependent effectiveness question.

- Reviewer Dtd3 (initial: 4):

  Would likely remain at 4. While the rebuttal addressed deployment framing and provided PPL reproducibility claims, the fundamental concern about marginal pipeline gains versus adaptive allocation remains. The baseline anomalies, though explained, reduce confidence in comparative claims. The reviewer's assessment that "the primary gain appears to come from adaptive allocation, not the sequential pipeline" is not convincingly refuted.

- Reviewer xhzF (initial: 6):

  May have increased to 8 as this reviewer had the most positive initial assessment and their specific questions were well addressed with new experiments. However, they may have found agreement with some of the additional concerns by other reviewers and decide to maintain their initial score.

---

### Decision · Program_Chairs · 2026-01-26

Reject